# Dyshomeostatic modulation of Ca²⁺-activated K⁺ channels in a human neuronal model of KCNQ2 encephalopathy

Dina Simkin[1,2], Kelly A Marshall[1], Carlos G Vanoye[2], Reshma R Desai[2], Bernabe I Bustos[1], Brandon N Piyevsky[1], Juan A Ortega[1], Marc Forrest[3,4], Gabriella L Robertson[1], Peter Penzes[3,4], Linda C Laux[5], Steven J Lubbe[1], John J Millichap[5], Alfred L George Jr[2]*, Evangelos Kiskinis[1,3]*

[1]The Ken & Ruth Davee Department of Neurology, Feinberg School of Medicine, Northwestern University, Chicago, United States; [2]Department of Pharmacology, Feinberg School of Medicine, Northwestern University, Chicago, United States; [3]Department of Physiology, Feinberg School of Medicine, Northwestern University, Chicago, United States; [4]Center for Autism and Neurodevelopment, Feinberg School of Medicine, Northwestern University, Chicago, United States; [5]Epilepsy Center and Division of Neurology, Departments of Pediatrics and Neurology, Ann & Robert H. Lurie Children's Hospital of Chicago, Feinberg School of Medicine, Northwestern University, Chicago, United States

*For correspondence:
al.george@northwestern.edu (ALGJ);
evangelos.kiskinis@northwestern.edu (EK)

Competing interests: The authors declare that no competing interests exist.

**Abstract** Mutations in *KCNQ2*, which encodes a pore-forming K⁺ channel subunit responsible for neuronal M-current, cause neonatal epileptic encephalopathy, a complex disorder presenting with severe early-onset seizures and impaired neurodevelopment. The condition is exceptionally difficult to treat, partially because the effects of *KCNQ2* mutations on the development and function of human neurons are unknown. Here, we used induced pluripotent stem cells (iPSCs) and gene editing to establish a disease model and measured the functional properties of differentiated excitatory neurons. We find that patient iPSC-derived neurons exhibit faster action potential repolarization, larger post-burst afterhyperpolarization and a functional enhancement of Ca²⁺-activated K⁺ channels. These properties, which can be recapitulated by chronic inhibition of M-current in control neurons, facilitate a burst-suppression firing pattern that is reminiscent of the interictal electroencephalography pattern in patients. Our findings suggest that dyshomeostatic mechanisms compound KCNQ2 loss-of-function leading to alterations in the neurodevelopmental trajectory of patient iPSC-derived neurons.

## Introduction

The *KCNQ2* gene encodes K$_V$7.2 (referred to here as KCNQ2), voltage-dependent potassium (K⁺) channels widely distributed in central and peripheral neurons. In mature neurons, KCNQ2 and the paralogous KCNQ3 protein form heterotetrameric channels (KCNQ2/3; *Wang et al., 1998*; *Schwake et al., 2006*). Together these channels mediate the M-current, a slowly activating and non-inactivating voltage-dependent K⁺ conductance suppressed by Gq protein-coupled muscarinic acetylcholine receptor activation (*Brown and Adams, 1980*). The M-current activates as neurons approach action potential (AP) threshold and acts to dampen neuronal excitability (*Brown and Adams, 1980*; *Brown and Passmore, 2009*). Therefore, KCNQ2/3 channels help set the AP

threshold and contribute to the post-burst afterhyperpolarization (AHP), which limits repetitive firing following bursts of action potentials (*Storm, 1989*; *Zhang and McBain, 1995*; *Devaux et al., 2004*; *Tzingounis and Nicoll, 2008*). These channels are enriched at the axon initial segment (AIS) and nodes of Ranvier, and are also expressed at lower densities at the soma, dendrites and synaptic terminals of neurons (*Devaux et al., 2004*; *Chung et al., 2006*; *Pan et al., 2006*; *Shah et al., 2008*).

The importance of KCNQ2 in normal brain development and function is underscored by genetic epilepsies associated with mutation of this channel. Disorders caused by *KCNQ2* mutations include benign familial neonatal seizures (BFNS), characterized by seizures that remit within the first year of life (*Charlier et al., 1998*; *Singh et al., 1998*) and the more severe neonatal developmental and epileptic encephalopathy (DEE; *Millichap and Cooper, 2012*; *Saitsu et al., 2012*; *Weckhuysen et al., 2012*; *Kato et al., 2013*). A search for *KCNQ2* on ClinVar, which tracks genetic variation in relation to disease, results in 710 different variants, some of which are recurrent. Mutations in *KCNQ2* account for approximately 5% of all mutations identified in genetic epilepsy (*von Deimling et al., 2017*; *Wang et al., 2017*) and 10% of those associated with early-onset forms of DEE (*Afawi et al., 2016*). The main clinical features of DEE are developmental and cognitive disabilities, with early onset of severe seizures that occur within a few days after birth, and are often refractory to antiepileptic drug treatment (*Auvin et al., 2016*).

The earliest hypothesis to explain epilepsy associated with *KCNQ2* mutations posited that loss of KCNQ2 channel function allows for sustained membrane depolarization after a single action potential leading to increased repetitive firing within bursts in excitatory neurons (*Cooper and Jan, 2003*). However, some variants associated with severe clinical phenotypes produce gain-of-function effects (*Millichap et al., 2017*; *Mulkey et al., 2017*). Enhanced $K^+$ conductance, specifically in the AIS, could hyperpolarize the AIS membrane and relieve steady state inactivation of sodium channels. This would increase the rate of action potential activation and action potential repolarization (*Niday and Tzingounis, 2018*).

The mechanisms by which developmental expression of KCNQ2 channels impact neuronal excitability are not clear. What remains elusive is how the defects in M-current affect the electrophysiological properties of human neurons leading to impaired neurodevelopment. Moreover, it is unclear whether KCNQ2-DEE pathogenesis results simply from altered M-channel function or from maladaptive cellular reorganization to compensate for chronic KCNQ2 channel dysfunction. The use of patient-specific induced pluripotent stem cell (iPSC) technology has enabled a new approach for elucidating pathogenic mechanisms of genetic disorders such as epileptic channelopathies, as it allows for the generation of otherwise inaccessible human neurons (*Ichida and Kiskinis, 2015*; *Mertens et al., 2016*; *Tchieu et al., 2017*; *Simkin and Kiskinis, 2018*). Here, we use KCNQ2-DEE patient-specific and isogenic control iPSC-derived excitatory neurons to elucidate the dynamic functional effects of a *KCNQ2* mutation during differentiation and maturation in culture.

## Results

### Establishing a human neuron model of KCNQ2 epileptic encephalopathy

To investigate the effects of mutant *KCNQ2* in human neurons, we isolated peripheral blood mononuclear cells (PBMCs) from a 13-year-old female clinically diagnosed with KCNQ2-DEE. Clinical genetic sequencing revealed a de novo heterozygous KCNQ2 variant (c.1742G>A) resulting in an arginine to glutamine missense mutation at position 581 (p.Arg581Gln, R581Q; KCNQ2 splice variant 1: NM_172107.4; *Figure 1A*). The arginine residue is highly conserved across species, and this mutation has been observed in at least four other individuals diagnosed with KCNQ2-DEE (ClinVar variation #21769; *Olson et al., 2017*; *Clinical Study Group et al., 2017*). Moreover, mutation of this residue to other amino acids has been reported in at least five other cases with similar pathology (R581L, R581P, and R581G; ClinVar variations #205920, #804957 and #265380, respectively), as well as in three individuals with less severe BFNS (R581Ter; ClinVar variation #21768). R581Q is located in a highly conserved coiled-coil domain of the C-terminus referred to as the C-helix. Disruption of this coiled-coil domain abolishes the ability of KCNQ2 to form functional homomeric and heteromeric (KCNQ2/KCNQ3) channels and prevents transport to the plasma membrane (*Schwake et al., 2006*). Thus, it is possible that R581Q precludes channel conductance by disrupting tetramerization and

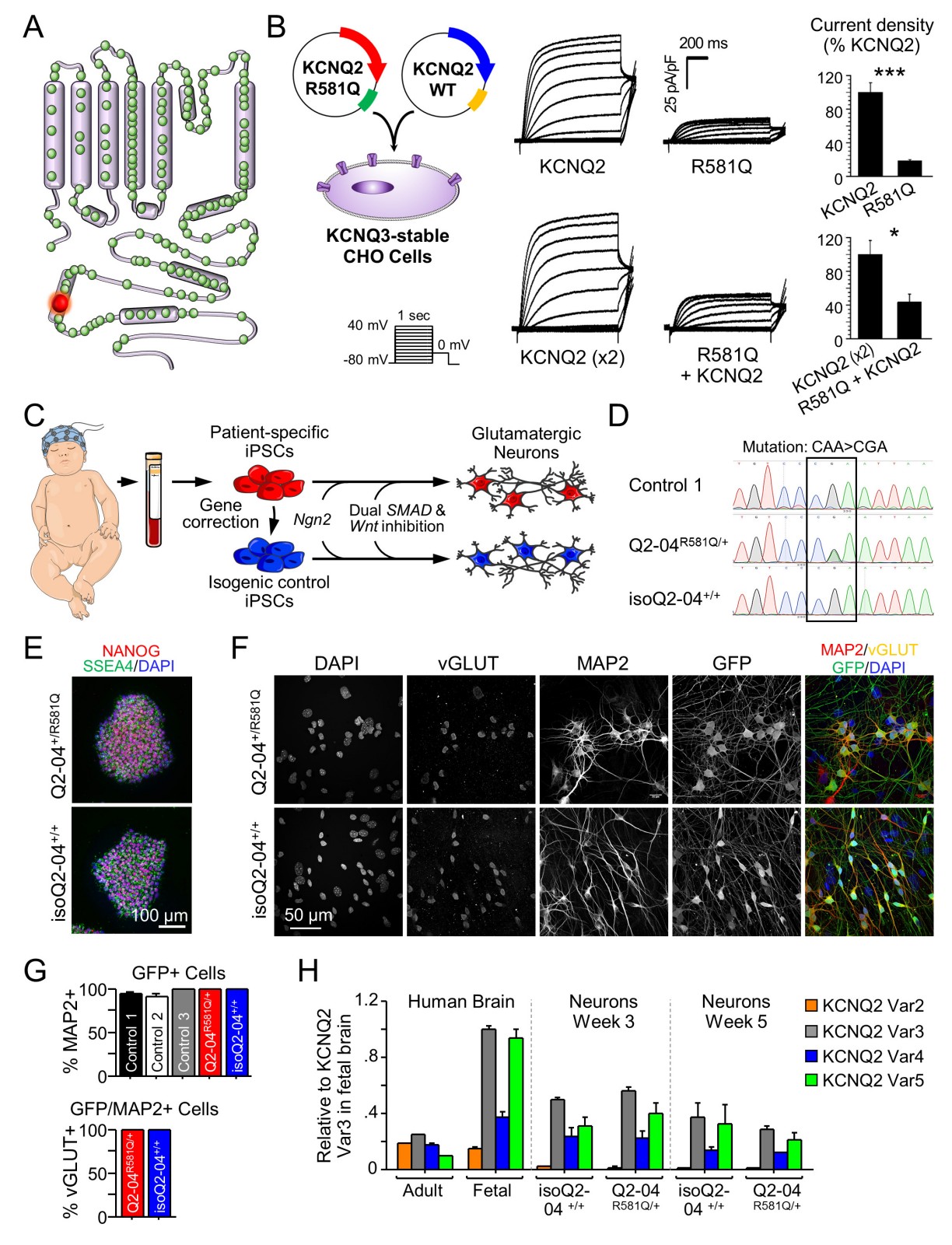

**Figure 1.** Generation of KCNQ2-DEE patient-specific iPSC-derived neurons. (**A**) Illustration of proposed structure of KCNQ2 channel subunit containing the mutation R581Q at the C-terminus (red) and other variants associated with KCNQ2-epileptic encephalopathy reported in ClinVar and (**Goto et al., 2019**) (green). (**B**) Heterologous expression of KCNQ2-R581Q. Left: transfection strategy and voltage pulse step protocol. Middle: Average XE-991-sensitive whole-cell currents normalized by membrane capacitance recorded using automated patch-clamp. KCNQ3-expressing CHO-K1 cells were

*Figure 1 continued on next page*

*Figure 1 continued*

transiently transfected with wild-type KCNQ2 (15 µg) or R581Q variant (15 µg) to recapitulate a homozygous state (top) or with R581Q (10 µg) plus wild-type KCNQ2 (10 µg) or wild-type KCNQ2 (x2; 20 µg) to mimic the heterozygous state (bottom). Right: Summary data (mean ± SEM) for average current density measured at +30 mV expressed as % of KCNQ2 WT values (KCNQ2: n = 58, R581Q: n = 63; KCNQ2 (x2): n = 21, R581Q + KCNQ2: n = 22). R581Q alone or combined with wild-type KCNQ2 produced 81.6 ± 10.7% (t test: ***p<0.0001) and 56.6 ± 14.9% (t test: *p=0.006) smaller current density, respectively, as compared to cells expressing wild-type channels. (C) Illustration of iPSC-derived cortical excitatory neuron platform. (D) DNA sequence electropherograms of *KCNQ2* in control and patient iPSCs before and after gene editing, demonstrate the correction of the heterozygous (R581Q; c.1742G>A) mutation (See *Figure 1—figure supplements 2* and *3* and *Supplementary files 1–3*). (E) Immunocytochemical labeling of KCNQ2-DEE patient-derived (Q2-04$^{R581Q/+}$) and isogenic control (isoQ2-04$^{+/+}$) iPSC lines with the pluripotency markers NANOG, SSEA4, and DAPI merged. Scale bar: 100 µm. (F) Immunocytochemical labeling with glutamatergic and neuronal markers vGLUT1 and MAP2 and GFP. Scale bar: 50 µm. (G) Quantification of GFP fluorescence coincident with MAP2 and vGLUT1 immuno-positive staining in three unrelated healthy controls and patient and isogenic control iPSC-derived neurons (See *Figure 1—figure supplement 4*). (H) RT-qPCR expression analysis of *KCNQ2* splice variants in the differentiated neuronal cultures on weeks 3 and 5 using isoform-specific primers (See *Supplementary file 4*). All values are normalized to fetal brain *KCNQ2* splice variant 3, as it is the highest expressing variant in all samples. Data from human adult and fetal brain are shown for comparison. The online version of this article includes the following source data and figure supplement(s) for figure 1:

**Source data 1.** Quantification of RT-qPCR expression ΔΔCt values of *KCNQ2* splice variants for *Figure 1H*.
**Figure supplement 1.** Whole-cell voltage-clamp analysis of KCNQ2 R581Q.
**Figure supplement 2.** Quality control studies of iPSC lines.
**Figure supplement 3.** CRISPR off-target and whole genome sequencing analysis of iPSC lines.
**Figure supplement 4.** Quality control studies of iPSC-derived neurons.

trafficking of the channel complex in neurons. However, this KCNQ2 C-terminus mutation has not previously been characterized functionally.

To determine the effect of KCNQ2-R581Q on channel function, CHO cells that stably expressed KCNQ3 were transiently transfected with KCNQ2 containing the KCNQ2-R581Q mutation alone (homozygous state) or in a 1:1 ratio with wild-type KCNQ2 (heterozygous state). We found that KCNQ2-R581Q alone or combined with wildtype KCNQ2 produced 81.6% (t test: p<0.0001) and 56.6% (t test: p=0.006) smaller current density, respectively, as compared to cells expressing wild-type channels (*Figure 1B* and *Figure 1—figure supplement 1*). These findings are consistent with a KCNQ2 loss-of-function.

We derived iPSCs from PBMCs using integration-free, Sendai virus-mediated reprogramming. To create a model that would allow us to attribute any phenotypic differences to the disease-associated genetic variant, we generated an isogenic mutation-corrected control iPSC line from the patient-derived line (*Figure 1C*). We specifically corrected the mutant allele using CRISPR/Cas9 genome editing and simultaneously introduced two silent mutations in the protospacer adjacent motif (PAM) to prevent re-cleavage (*Figure 1—figure supplement 2A–C*). We validated the presence of the heterozygous mutation in the patient-derived iPSCs (Q2-04$^{R581Q/+}$) and identified a mutation-corrected isogenic clonal cell line (isoQ2-04$^{+/+}$) by targeted PCR and Sanger sequencing (*Figure 1D*). The resulting iPSCs exhibited a normal female karyotype, typical stem cell morphology, and expressed pluripotency markers, including nuclear NANOG and the cell surface antigen SSEA4 (*Figure 1E* and *Figure 1—figure supplement 2D–E*). We found no evidence for off-target edits in any of the top 10 predicted genomic regions with homology to the targeted *KCNQ2* exon using a T7 endonuclease-based assay (*Figure 1—figure supplement 3A* and *Supplementary file 1* and *2*). As an additional quality control, we performed whole genome sequencing (WGS) of samples Q2-04$^{R581Q/+}$ (29.41x coverage) and isoQ2-04$^{+/+}$ (28.12x coverage) and determined that they correspond to the same individual (pi_hat score = 0.99). We also validated the genetic correction and integrity of the targeted gene (*Figure 1—figure supplement 2C*), and found that there were no edits in any of all 74 predicted off-target sites indicating highly specific CRISPR activity (*Figure 1—figure supplement 3B*, inner arrows, *Supplementary file 3*).

Given the clinical presentation of KCNQ2-associated DEE and the focal source of seizures that reside in the cortex (*Wilmshurst et al., 2015*; *Fisher et al., 2017*), we chose to study cortical excitatory neurons, differentiated through a modified *NGN2* overexpression protocol (*Figure 1—figure supplement 4A*; *Zhang et al., 2013*; *Nehme et al., 2018*). We simultaneously differentiated the Q2-04$^{R581Q/+}$ patient-derived line, the engineered isogenic control line (isoQ2-04$^{+/+}$) and three iPSC lines generated from unrelated, healthy control individuals (*Figure 1—figure supplement 4B*;

*Boulting et al., 2011*). Expression of GFP marked the lentivirus-transduced cells that were co-cultured with primary mouse glia to facilitate in vitro neuronal maturation. To determine the efficiency of differentiation, we used immunocytochemistry (ICC) to quantify the percentage of MAP2, vGLUT1, and GFP-positive cells (*Figure 1F,G*). Over 95% of GFP-positive cells were also MAP2-positive for all iPSC lines. Furthermore, all neurons that were GFP and MAP2 positive also expressed vGLUT1 (*Figure 1F,G* and *Figure 1—figure supplement 4C*). As previously described (*Zhang et al., 2013*), we found that these cultures expressed high levels of *vGLUT2, FOXG1*, and *BRN2*, which are characteristic of excitatory layer 2/3 cortical neurons (*Figure 1—figure supplement 4D* and *Supplementary file 4*). Importantly, we confirmed that the differentiated neuronal cultures expressed several *KCNQ2* splice variants by RT-qPCR (*Figure 1H*).

## KCNQ2-DEE neurons exhibit enhanced spontaneous phasic bursting

One of the clinical hallmarks of KCNQ2-DEE is an interictal burst-suppression pattern (i.e. paroxysmal bursts of activity interspersed with periods of electrical silence) recorded by EEG during the first few days of life (*Weckhuysen et al., 2012*; *Milh et al., 2013*; *Millichap et al., 2016*). The *KCNQ2* iPSC-based platform that we developed presents an opportunity to dissect the functional consequences of KCNQ2 mutations during neuronal development in vitro. To assess spontaneous neuronal activity, we used multi-electrode arrays (MEAs) and performed daily recordings over a 3-week period (days 15–31, N = 3 independent differentiations; *Figure 2A,B*). We plated an equal number of Q2-04$^{R581Q/+}$ and isoQ2-04$^{+/+}$ neurons and carefully monitored neuronal attachment throughout the time course of experiments (p=0.541; *Figure 2A,C*). As neurons matured, neuronal cultures acquired significant spontaneous activity ($\geq$10% of the electrodes active) on or after day 15 (*Figure 2—figure supplement 1A*). Interestingly, Q2-04$^{R581Q/+}$ neurons exhibited a significantly greater number of active electrodes (p<0.0001) and slightly higher spontaneous firing frequency (p=0.0081) early in their maturation (DIV: 10-13), relative to isoQ2-04$^{+/+}$ neurons (*Figure 2—figure supplement 1A,B*). This suggests that Q2-04$^{R581Q/+}$ neurons are able to fire spontaneously earlier than isoQ2-04$^{+/+}$ neurons. However, the number of active electrodes was not different between Q2-04$^{R581Q/+}$ and isoQ2-04$^{+/+}$ neurons at later times in culture (DIV: 15-31) (p=0.0896; *Figure 2—figure supplement 1A*). More strikingly, the distribution of spiking in Q2-04$^{R581Q/+}$ neurons was dramatically and increasingly irregular compared to isogenic control neurons, indicated by the increased inter-spike interval coefficient of variation (ISI CoV; p<0.0001; *Figure 2—figure supplement 1C*). This irregularity was associated with greater bursting propensity, with short intervals between spikes in bursts and longer intervals between bursts.

While firing frequency was marginally higher in Q2-04$^{R581Q/+}$ neurons (p=0.0205; *Figure 2—figure supplement 1B*), mere hyperexcitability may not be an accurate recapitulation of the developmental deficits in epileptic encephalopathy (*Simkin and Kiskinis, 2018*). Thus, we focused on the neuronal firing pattern over time in culture rather than individual time points of higher firing frequency. As illustrated by representative spike raster plots, compared to isoQ2-04$^{+/+}$ neurons, Q2-04$^{R581Q/+}$ neuronal cultures fired significantly more bursts (p<0.0001) and had a greater number of bursting electrodes (p<0.0001) that increased in number faster, as neurons matured over time (repeated measures ANOVA interaction: p<0.0001, p=0.0052, respectively; *Figure 2D* and *Figure 2—figure supplement 1D*). Furthermore, their spikes were more restricted to bursts, as demonstrated by significantly higher ISI CoV (p<0.0001), number of spikes per burst (p<0.0001), percentage of all spikes that occurred within bursts (burst %; p<0.0001) and burst frequency (p=0.0072) as compared to isogenic control neurons (*Figure 2D,E* and *Figure 2—figure supplement 1C–E*). These results indicate that Q2-04$^{R581Q/+}$ neurons were more prone to fire in bursts rather than single tonic spikes compared to isogenic control neurons (*Figure 2E*). Moreover, the significant interaction of genotype and day in culture suggest that this bursting phenotype became even more pronounced and phasic over time. This is further supported by the fact that Q2-04$^{R581Q/+}$ neurons developed significantly smaller inter-burst interval CoV only after day 22 in culture, and thus their bursts became more regular and phasic (p=0.0169; *Figure 2—figure supplement 1F*).

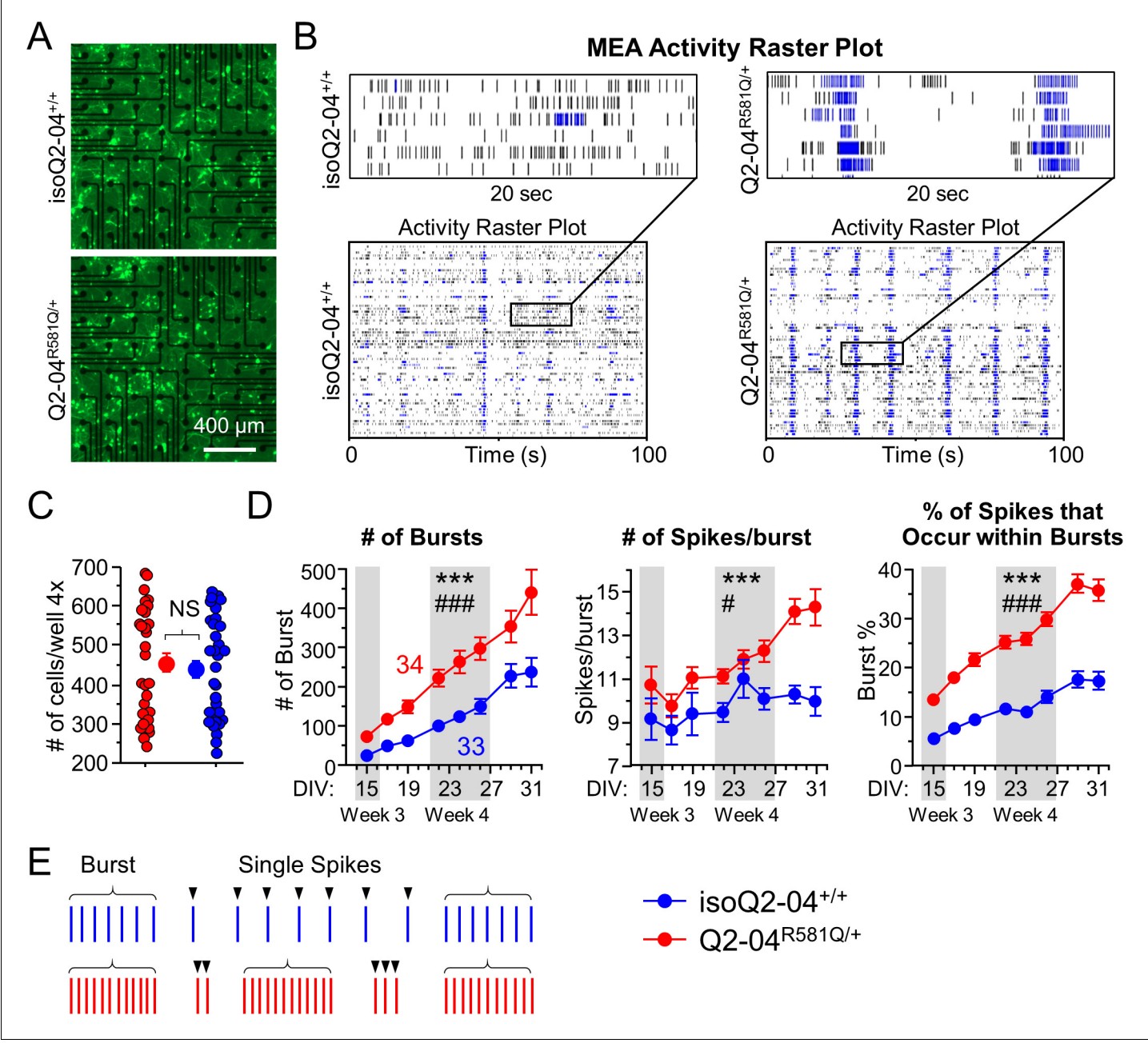

**Figure 2.** KCNQ2-DEE neurons exhibit enhanced spontaneous phasic bursting. (**A**) Representative images at ×4 magnification of GFP-fluorescing KCNQ2-DEE and isogenic control neurons plated on MEA wells on day 24 in culture. Scale bar: 400 µm (**B**) Representative spike raster plots from a single MEA well of KCNQ2-DEE (Q2-04$^{R581Q/+}$) and isogenic control (isoQ2-04$^{+/+}$) neurons on day 31. Each row represents the signal detected by a single electrode; black ticks indicate single spikes and blue ticks spikes that occur within bursts. (**C**) Each MEA plate well was imaged on day 24 in culture using ×4 magnification. GFP-fluorescing neurons on the electrode field area were counted for each well of every plate. The average number of cells per well was not different between Q2-04$^{R581Q/+}$ (459.2 ± 24.3 neurons/well) and isoQ2-04$^{+/+}$ (438.8 ± 22.5 neurons/well) neurons (t test: p=0.5410, N = 34 and 33 wells, respectively). (**D**) Longitudinal analysis of MEA recordings from days 15 to 31 in culture. Compared to isogenic control neurons, Q2-04$^{R581Q/+}$ neurons had increased average number of bursts detected on single electrodes (repeated measures ANOVA for genotype: $F_{(1,455)}$ = 17.31, ***p<0.0001; genotype/day interaction: $F_{(7,455)}$ = 3.88, ###p=0.0004); number of spikes within bursts (repeated measures ANOVA for genotype: $F_{(1,455)}$ = 17.31, ***p<0.0001; genotype/day interaction: $F_{(7,455)}$ = 2.24, #p=0.0301); and percentage of all detected spikes which were found to occur within bursts (repeated measures ANOVA for genotype: $F_{(1,455)}$ = 135.47, ***p<0.0001; genotype/day interaction: $F_{(7,455)}$ = 5.89, ###p<0.0001; See *Figure 2—figure supplement 1*). (**E**) Illustration of firing pattern showing increased phasic firing in bursts of Q2-04$^{R581Q/+}$ neurons (red) as compared to isogenic control neurons (blue). Number of wells analyzed per cell line is displayed within the figure from three independent differentiations. Values displayed are mean ± SEM.

*Figure 2 continued on next page*

*Figure 2 continued*

The online version of this article includes the following source data and figure supplement(s) for figure 2:

**Source data 1.** Quantification of DIV 15–31 MEA data parameters for *Figure 2* and *Figure 2—figure supplement 1*.
**Figure supplement 1.** MEA quality control and bursting measurements.
**Figure supplement 1—source data 1.** Quantification of DIV 10–13 MEA data parameters for *Figure 2—figure supplement 1A and B*.

## KCNQ2-DEE neurons exhibit enhanced AP repolarization and Post-Burst AHP

The altered pattern of firing activity suggested differences in the intrinsic excitability properties of Q2-04$^{R581Q/+}$ neurons. To determine the source of the progressive increase in bursting in KCNQ2-DEE neurons, we performed whole-cell current-clamp measurements and examined the AP properties and post-burst AHPs. We systematically recorded from single Q2-04$^{R581Q/+}$ and isogenic control GFP-positive cortical neurons at three time points in culture defined as week 3 (DIV: 14–16), week 4 (DIV: 22–26), and week 5 (DIV:32–35; *Figure 3A,B*). As neurons matured in culture, the neuronal resting membrane potentials (RMP) became more hyperpolarized and AP amplitudes increased for both genotypes. However, Q2-04$^{R581Q/+}$ neurons had a significantly more depolarized RMP (p=0.0163) and higher input resistance (p=0.0075) only during week 3 (*Figure 3—figure supplement 1A* and *Table 1*). This could be a result of the reduced M-current or a change in the way Q2-04$^{R581Q/+}$ neurons mature. The increasing AP amplitudes for both genotypes over the course of 5 weeks indicate some progress in functional neuronal maturation (*Figure 3—figure supplement 1B* and *Table 1*; *Bardy et al., 2016*; *Nehme et al., 2018*; *Lindhout et al., 2020*).

We observed that compared with isoQ2-04$^{+/+}$ neurons, Q2-04$^{R581Q/+}$ neurons had slower AP repolarization with longer AP half-width at week 3 (p=0.0383; *Figure 3C,D*; *Table 1*). Slower AP repolarization is consistent with a reduced M-current. However, Q2-04$^{R581Q/+}$ neurons progressively developed faster AP repolarization with shorter AP half-widths at week 5 and larger fast component of the AHP (fAHP) at week 4 (p=0.03) and at week 5 (p=0.035; *Figure 3C,D* and *Table 1*). There was no difference in AP thresholds. Faster AP repolarization would enable neurons to fire a greater number of APs more rapidly with less synaptic input, which may explain the enhanced number of spikes per burst and higher tendency of Q2-04$^{R581Q/+}$ neurons to fire within bursts (*Figure 2D*).

The ability of neurons to fire in bursts is facilitated by an increased AP generation capacity and by the refractory period after each period of high frequency spiking. Several K$^+$ conductances activate following a burst of APs to hyperpolarize the membrane, restore ionic balance and prevent neurons from firing. Previous studies have shown the involvement of KCNQ2 channels in the medium AHP (mAHP) in cortical neurons (*Guan et al., 2011*; *Battefeld et al., 2014*). We examined the post-burst AHP using a 50 Hz train of 25 APs evoked by 2 ms/1.4 nA current pulses. Compared to controls, Q2-04$^{R581Q/+}$ neurons exhibited significantly larger mAHPs (peak of AHP) and slow post-burst AHPs (sAHP; 1 s after last stimulus; p<0.0001 and p<0.0001, respectively; *Figure 3E,F* and *Table 1*). Surprisingly, isogenic control neurons exhibited a time-dependent attenuation in the mAHP and sAHP and thus were significantly smaller than Q2-04$^{R581Q/+}$ neurons during weeks 4–5 but not earlier (*Figure 3E,F* and *Table 1*).

Although an enhanced AHP increases the refractory latency of neurons to repolarize and fire again after a burst of APs, the faster repolarization observed for Q2-04$^{R581Q/+}$ neurons may counterbalance this effect. These findings are consistent with enhanced phasic burst firing in Q2-04$^{R581Q/+}$ neurons. The changes in AP properties account for more APs fired within bursts, whereas enhanced post-burst AHPs dampen spontaneous firing between bursts (*Figure 2*).

## Acute inhibition of M-current impairs AP repolarization and post-burst AHP

Previous studies have shown that blocking M-current in rodent cortical excitatory neurons acutely with XE991 enhances neuronal excitability by lowering AP threshold, impairing AP repolarization and attenuating the post-burst AHP (*Yue and Yaari, 2004*; *Santini and Porter, 2010*; *Lezmy et al., 2017*). However, our analysis of Q2-04$^{R581Q/+}$ neurons demonstrated that AP repolarization was slower during week 3 (with longer AP half-width) compared to isoQ2-04$^{+/+}$, but over time AP repolarization became more pronounced with faster AP half-width and enhanced fAHP. Furthermore, the

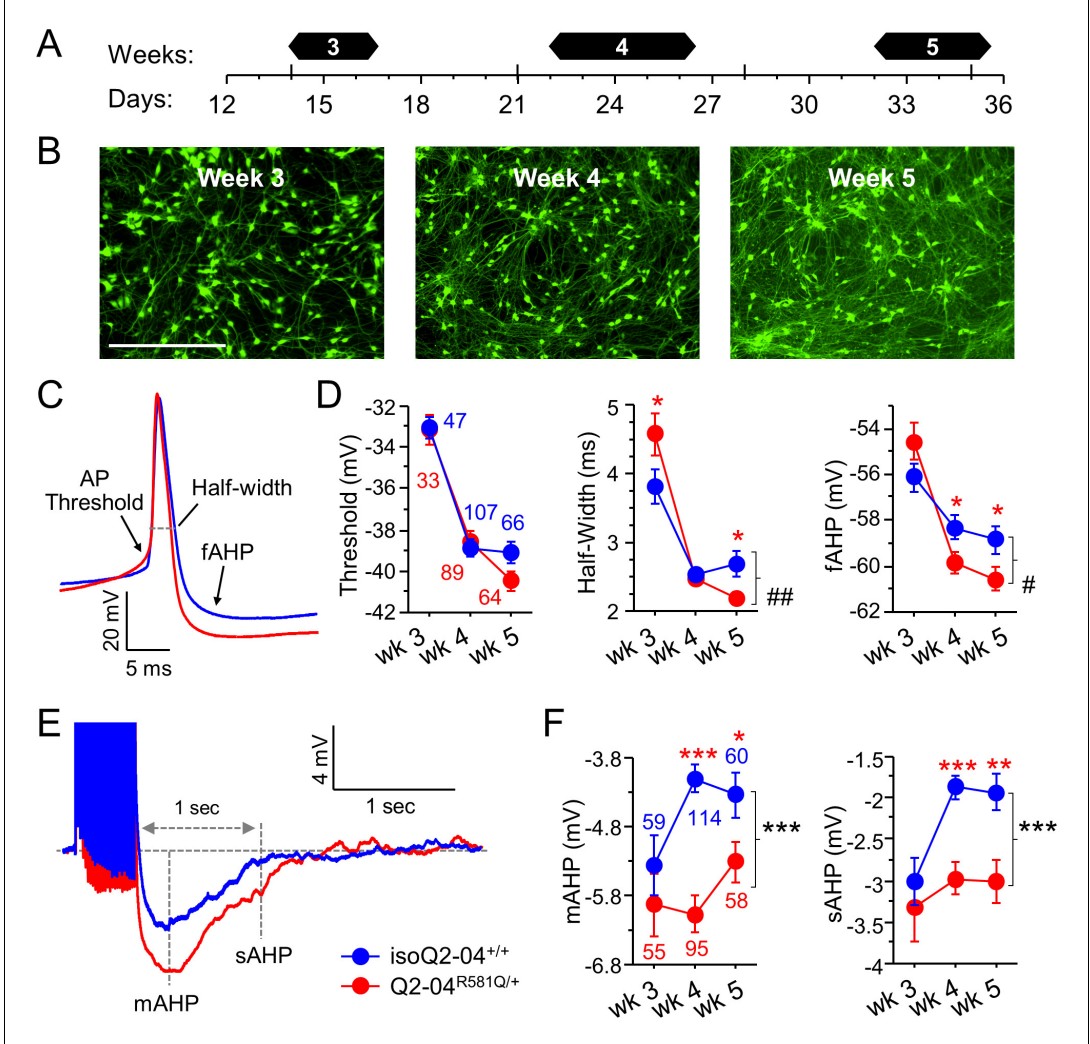

**Figure 3.** KCNQ2-DEE neurons exhibit progressive enhancement of AP repolarization and post-burst AHP. (**A**) Experimental time line. (**B**) Representative images of GFP-fluorescing isogenic control neurons during weeks 3, 4, and 5. Scale bar: 400 μm. (**C**) Representative whole-cell current-clamp traces showing AP amplitude, threshold, half-width and fAHP at week 5. (**D**) Analysis of AP properties measured at weeks 3, 4, and 5. Q2-04R581Q/+ (red) neurons had no difference in AP threshold (two-way ANOVA for genotype: $F_{(1,400)}$=0.82; p=0.37) but exhibited a progressive enhancement in AP repolarization over time with shorter AP half-widths (two-way ANOVA for genotype/weeks interaction: $F_{(2,400)}$ = 7.2; ##p=0.0008) and larger fAHPs (two-way ANOVA for genotype/weeks interaction: $F_{(2,400)}$ = 3.5; #p=0.022). Posthoc analysis using t-tests to compare each time point revealed longer half-widths on week 3 (*p=0.0383) but then shorter half-widths by week 5 (*p=0.0206) in Q2-04R581Q/+ neurons. fAHP was larger in Q2-04R581Q/+ neurons only at week 4 (*p=0.0298) and 5 (*p=0.0349), but was not significantly smaller at week 3 (p=0.0805). (**E**) Representative traces showing post-burst AHPs after 50 Hz train of 25 APs evoked by 2 ms/1.2 nA suprathreshold current stimuli. (**F**) Q2-04R581Q/+ neurons had enhanced mAHP (two-way ANOVA for genotype: $F_{(1,435)}$ = 19.99; ***p<0.0001; genotype/weeks interaction: $F_{(2,435)}$=2.94; p=0.054) and sAHP (two-way ANOVA for genotype: $F_{(1,435)}$ = 18.42; ***p<0.0001; genotype/weeks interaction: $F_{(2,435)}$=1.31; p=0.271). Posthoc analysis using t-tests to compare Q2-04R581Q/+ and isogenic control neurons at each time point revealed differences in mAHP and sAHP only at weeks 4 and 5, with no significant differences at week 3. Number of neurons analyzed is displayed within the figure and in **Table 1** (Also see **Figure 3—figure supplement 1**). Red * indicate significance between Q2-04R581Q/+ and isoQ2-04+/+ neurons at each individual time point using posthoc t-tests. Values displayed are mean ± SEM.

The online version of this article includes the following source data and figure supplement(s) for figure 3:

**Source data 1.** Quantification of passive and active current-clamp parameters for **Figure 3** and **Figure 3—figure supplement 1A and B**.

**Figure supplement 1.** Intrinsic excitability passive and active properties and effects of acute XE991 application.

post-burst AHPs are large throughout the timeline of our experiments (**Figure 3**). We propose two potential explanations for these divergent mechanisms for altered excitability: either M-current inhibition in human cortical glutamatergic neurons has different effects than in rodent neurons, or

**Table 1.** Developmental timeline of intrinsic membrane properties of patient-derived and isogenic control neurons.

| Weeks: | Week 3 | | Week 4 | | Week 5 | |
|---|---|---|---|---|---|---|
| Genotype: | isoQ2-04[+/+] | Q2-04[R581Q/+] | isoQ2-04[+/+] | Q2-04[R581Q/+] | isoQ2-04[+/+] | Q2-04[R581Q/+] |
| Resting Potential (mV) | −53.5 ± 0.7 (74) | −51.3 ± 0.6 * (72) | −54.1 ± 0.5 (143) | −55.2 ± 0.6 (137) | −55.8 ± 0.6 (71) | −56.7 ± 0.7 (73) |
| Input Resistance $R_N$ at RMP (MΩ) | 900.6 ± 47 | 1079.9 ± 46.5 * | 845.2 ± 27.9 | 850.3 ± 25.7 | 871.5 ± 29.4 | 816.3 ± 27.9 |
| Series Resistance $R_S$ RMP (MΩ) | 14.5 ± 0.29 | 15 ± 0.3 | 14.4 ± 0.3 | 14.1 ± 0.3 | 15.6 ± 0.3 | 15.5 ± 0.4 |
| AP Amplitude from Baseline (mV) | 88 ± 0.7 (47) | 87.6 ± 1 (33) | 91.1 ± 0.6 (107) | 92.6 ± 0.6 (89) | 90.6 ± 0.8 (66) | 92 ± 0.9 (64) |
| AP Threshold (mV) | −33 ± 0.5 | −33.3 ± 0.7 | −38.8 ± 0.4 | −38.6 ± 0.5 | −39.1 ± 0.5 | −40.4 ± 0.5 |
| AP Half-Width (ms) | 3.8 ± 0.3 | 4.6 ± 0.3 * | 2.5 ± 0.1 | 2.5 ± 0.1 | 2.7 ± 0.2 | 2.2 ± 0.1 * |
| fAHP (mV) | −56.1 ± 0.6 | −54.5 ± 0.7 | −58.2 ± 0.5 | −59.8 ± 0.5* | −58.8 ± 0.6 | −60.6 ± 0.5 * |
| mAHP (mV) | −5.3 ± 0.4 (59) | −5.9 ± 0.4 (55) | −4.1 ± 0.2 (114) | −6 ± 0.3 *** (95) | −4.3 ± 0.3 (60) | −5.2 ± 0.3 * (58) |
| sAHP (mV) | −3 ± 0.3 | −3.4 ± 0.4 | −1.9 ± 0.1 | −3 ± 0.2 *** | −1.9 ± 0.2 | −3 ± 0.3 ** |

*p< 0.05, **p<0.005, ***p<0.0005: *posthoc* Fisher PLSD test comparing Q2-04[R581Q/+] neurons to isoQ2-04[+/+] isogenic controls during each week. Values displayed are Mean ± SEM. Number of neurons is indicated in ().

RMP - resting membrane potential; AP - Action potential; AHP - afterhyperpolarization; fAHP - fast AHP; mAHP - medium AHP; sAHP – slow post-burst AHP.

chronic M-current suppression leads to enhanced burst firing and altered AP properties by indirect mechanisms that are different from those related to acute M-current inhibition.

To investigate further, we interrogated the intrinsic membrane properties of unrelated control, week 4 excitatory neurons before and after acute treatment with 20 μM XE991. Consistent with previous reports (*Yue and Yaari, 2004*; *Santini and Porter, 2010*; *Lezmy et al., 2017*), XE991 significantly lowered AP threshold and slowed AP repolarization (AP threshold: p=0.0104; HW: p=0.0046; fAHP: p=0.009; *Figure 3—figure supplement 1C–E*). Furthermore, XE991 blunted the mAHP and sAHP amplitudes in these neurons (mAHP: p=0.0014; sAHP: p=0.022; *Figure 3—figure supplement 1F,G*). These effects did not resemble the behavior of Q2-04[R581Q/+] neurons, suggesting that loss of M-current alone is not sufficient for phasic burst firing.

Collectively, these experiments suggest that KCNQ2-DEE neurons develop a more pronounced phasic bursting phenotype progressively as a result of chronic M-current reduction coupled with dyshomeostatic adaptation of intrinsic membrane properties at different temporal scales.

## KCNQ2-DEE neurons exhibit altered K[+]channel gene expression

The intrinsic AP properties of Q2-04[R581Q/+] neurons are not consistent with a pure loss of M-current but rather with a gain of other fast voltage-gated and Ca[2+]-dependent K[+] conductances that increase over time in culture on different time scales. While a number of different channels might contribute to a dyshomeostatic mechanism, strong candidates are large, intermediate, and small conductance, Ca[2+]- and voltage-gated BK (*KCNMA1*) and IK/SK (*KCNN4/KCNN1-3*) channels that participate in fast AP repolarization and slow Ca[2+]-dependent post-burst AHP, respectively (*Storm, 1990*; *Iyer et al., 2017*; *Latorre et al., 2017*). The kinetics of BK channels are modulated by several β subunits transforming them from non-inactivating if alone or associated with β1 (*KCNMB1*) or β4 (*KCNMB4*), to fast inactivating if associated with β2 (*KCNMB2*) or β3 (*KCNMB3*) subunits (*Storm, 1989*; *Xia et al., 1999*; *Hu et al., 2003*; *Kaufmann et al., 2010*; *Li and Yan, 2016*; *Latorre et al., 2017*). To determine if any of these channels or their accessory subunits are upregulated progressively over time in Q2-04[R581Q/+] neurons, we examined the levels of gene expression using RT-qPCR on weeks 3 and 5. We also examined the gene expression levels of several other ion channels including *KCND2, KCNT1, KCNA1, KCNA2, KCNA4, KCNQ2, KCNQ3, KCNQ5, HCN1, HCN2, SCN8A*, and *ANKG* (known to bind KCNQ2/3 in the AIS). On week 3, we found no difference in expression of *KCNMB1-4, KCNN1-3, KCND2, KCNT1, KCNA1, KCNA2, KCNQ2, KCNQ3, KCNQ5, HCN1, HCN2, SCN8A*, and *ANKG*. By contrast, *KCNMA1* (BK channel), *KCNN4* (IK channel), and *KCNA4* (encoding K$_V$1.4 channel) expression was higher in Q2-04[R581Q/+] neurons (p=0.0429, p=0.016 and p=0.0216, respectively; *Figure 4A* and *Figure 4—figure supplement 1A*). By week 5, the difference in *KCNMA1, KCNN4* and *KCNA4* was no longer significant but other

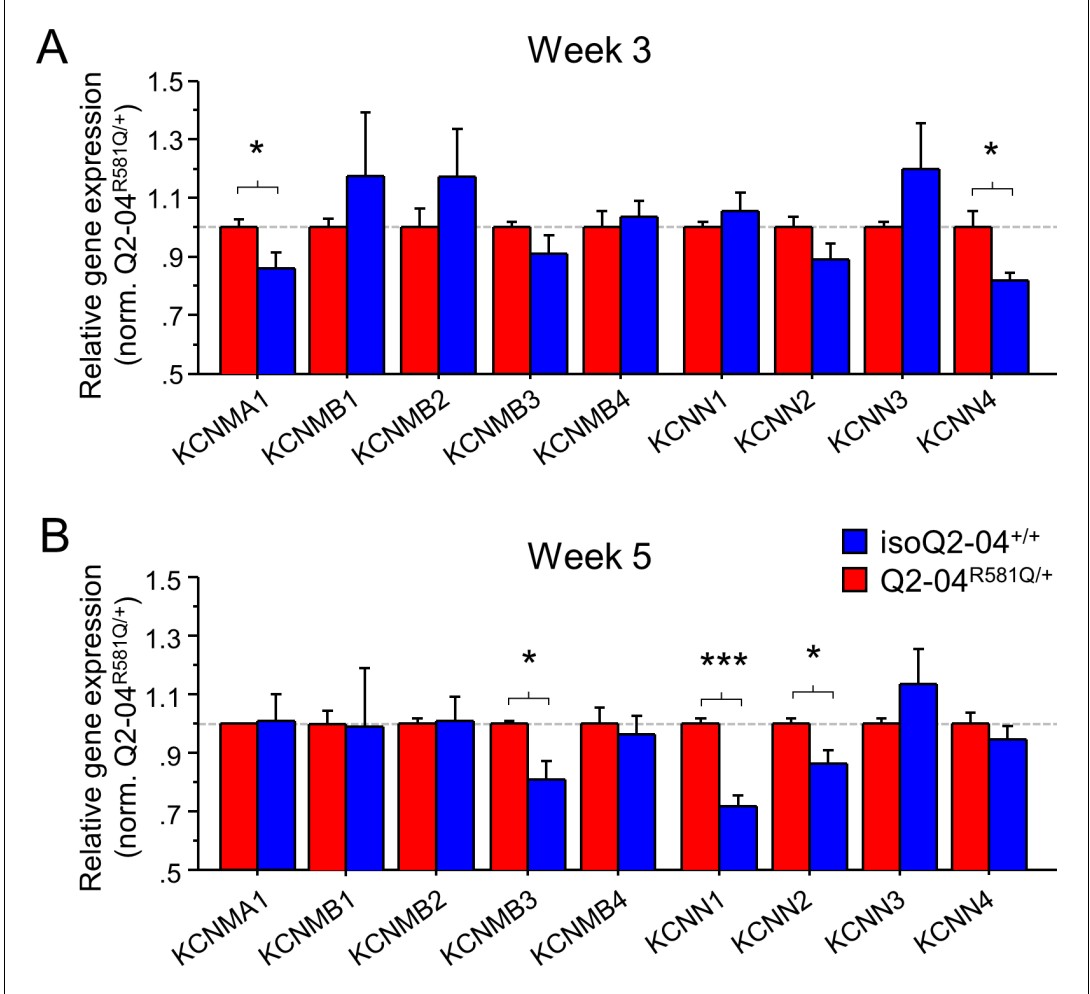

**Figure 4.** Enhanced expression of $Ca^{2+}$-activated $K^+$ channel genes in KCNQ2-DEE neurons. (**A**) qPCR gene expression pattern of $K^+$ channels and major β subunits involved in AP repolarization and post-burst AHP on week 3. *KCNMA1* and *KCNN4* expression was significantly higher in Q2-04$^{R581Q/+}$ neurons (t-test: *p=0.0429 and *p=0.016, respectively). (**B**) At week 5, expression of *KCNMB3*, *KCNN1* and *KCNN2* was significantly higher in Q2-04$^{R581Q/+}$ neurons (t test: *p=0.0081, ***p<0.0001 and *p=0.018, respectively). Values displayed are mean ± SEM from three independent differentiations normalized within each differentiation to Q2-04$^{R581Q/+}$ of each time-point.

The online version of this article includes the following source data and figure supplement(s) for figure 4:

**Source data 1.** Quantification of RT-qPCR expression ΔΔCt values for *Figure 4* and *Figure 4—figure supplement 1*.

**Figure supplement 1.** Expression of ion channel genes in KCNQ2-DEE neurons.

differences had emerged, including an upregulation of *KCNMB3*, *KCNN1*, *KCNN2*, *KCNA1*, and *KCNA2*, and downregulation of *KCNQ5* expression in Q2-04$^{R581Q/+}$ neurons relative to controls (p=0.0081, p<0.0001, p=0.018, p=0.0099, p=0.0017 and p=0.0198, respectively; *Figure 4B* and *Figure 4—figure supplement 1B*). These genes, which encode a BK channel β-subunit, SK channels (SK1 and SK2), $K_V1.1$ and $K_V1.2$, respectively, have been implicated in epilepsy and developmental delay in human and animal studies (*Hu et al., 2003*; *Lorenz et al., 2007*; *McKay et al., 2012*; *Lerche et al., 2013*; *Masnada et al., 2017*; *Paulhus et al., 2020*), and their upregulation is consistent with a gain of a $K^+$ conductance suggested by our electrophysiological studies (i.e. BK/SK upregulation can lead to larger post-burst AHPs and BK/$K_V1.1$/$K_V1.2$ upregulation can lead to faster AP repolarization).

## KCNQ2-DEE neurons exhibit a dyshomeostatic increase in BK and SK channel function

To determine if SK channels were functionally altered in KCNQ2-DEE neurons, we investigated their contribution to the post-burst AHP in Q2-04$^{R581Q/+}$ and isogenic control neurons at week 4. Addition of apamin (500 nM), a SK channel antagonist, reduced the mAHP in both groups, and reduced the sAHP only in Q2-04$^{R581Q/+}$ neurons. The magnitudes of change of the mAHP (ΔmAHP: p=0.0001) and sAHP (ΔsAHP: p<0.0001) were significantly larger for Q2-04$^{R581Q/+}$ neurons compared to isoQ2-04$^{+/+}$ neurons (*Figure 5A–C*). These data suggest that functional enhancement of SK channels in KCNQ2-DEE neurons contributes to the increase of slowly deactivating Ca$^{2+}$-activated currents that determine the level of post-burst AHP.

Furthermore, given the observed enhancement of AP repolarization at week 5, we applied apamin (500 nM) and paxilline (20 µM, a BK channel antagonist) during spontaneous MEA recording on day 32 in culture. Blocking SK and BK channels reduced the ISI CoV of Q2-04$^{R581Q/+}$ neurons (p=0.0015) to the level observed in untreated controls, while treatment did not affect isoQ2-04$^{+/+}$ neurons (p=0.796; *Figure 5D*). Addition of both drugs also had a significantly larger effect on the number of spikes/burst (ΔSpikes/Burst: p=0.0087) and burst % (ΔBurst%: p=0.0254) in Q2-04$^{R581Q/+}$ neurons than in isogenic control neurons. This pharmacological treatment specifically reduced the bursting features to the levels seen in isogenic control neurons before drug application (*Figure 5D*). Together, these data suggest that inhibition of SK and BK channels in Q2-04$^{R581Q/+}$ neurons reduces post-burst AHPs and normalizes phasic burst firing behavior.

## Chronic M-current inhibition in control neurons phenocopies KCNQ2-DEE

To determine if the adaptive enhancement of repolarization, post-burst AHP and increased bursting by enhanced K$^+$ channel functional expression is a result of early and sustained suppression of M-current, we chronically treated isogenic control neurons with a low concentration of XE991 (1 µM, starting on day 12 in culture; *Figure 6A*) and measured excitability during week 4. Chronically XE991-treated control neurons exhibited enhanced repolarization (half width: p=0.0002; fAHP: p=0.01) and larger post-burst AHPs (mAHP: p=0.0001; sAHP: p=0.0008) relative to untreated controls (*Figure 6B–E*, *Figure 6—figure supplement 1A,B* and *Table 2*). Importantly, these effects were identical to or larger than the properties observed for untreated Q2-04$^{R581Q/+}$ neurons. This experimental paradigm effectively phenocopied the electrophysiological behavior of Q2-04$^{R581Q/+}$ neurons in isoQ2-04$^{+/+}$ neurons and suggests that the enhanced AP repolarization and larger post-burst AHP that we identified in KCNQ2-DEE neurons, occur as a result of long-term reduction of M-current.

We next assessed the effects of chronic M-current inhibition on spontaneous neuronal activity using MEAs. Within 24 hr of adding XE991 to the neuronal media (DIV: 12-13), we observed an increase in the number of active electrodes approximately three times higher in XE991-treated relative to untreated isoQ2-04$^{+/+}$ neurons (p<0.0001; *Figure 6—figure supplement 2A,C*). This was accompanied by a sevenfold increase in burst % (p<0.0001; *Figure 6—figure supplement 2B,C*). However, the magnitude of change in mean firing frequency on active electrodes was not significantly affected by XE991 in isoQ2-04$^{+/+}$ neurons (p=0.4557; *Figure 6—figure supplement 2A*). Thus, XE991 induced a relatively fast upregulation of spontaneously firing isoQ2-04$^{+/+}$ neurons to the levels of Q2-04$^{R581Q/+}$ neurons (*Figure 6—figure supplement 2*). We did not find any difference in the average number of GFP-positive cells counted per well between the groups of neurons and at later time points the number of active electrodes became equal between the groups (*Figure 6—figure supplement 1C,D*). Furthermore, chronic M-current inhibition led to long-term enhanced burst firing parameters such as ISI CoV, number of bursting electrodes and bursts, number of spikes per burst and burst % (p<0.0001, p=0.0011, p=0.0275, p<0.0001, p<0.0001, respectively; *Figure 6F* and *Figure 6—figure supplement 1C*). The effects of chronic M-current inhibition on bursting in control neurons were more dramatic than the inherent bursting phenotype in Q2-04$^{R581Q/+}$ neurons, likely because of the less severe loss of function associated with the mutation relative to the pharmacological treatment with XE991 (*Figure 1B*).

Additionally, we compared the expression of K$^+$ channel genes involved in AP repolarization and post-burst AHP between Q2-04$^{R581Q/+}$, isoQ2-04$^{+/+}$ and isoQ2-04$^{+/+}$ that were chronically treated with XE991 at week 5. Expression of *KCNMA1*, *KCNMB3*, *KCNN1*, *KCNN2*, *KCNN4*, *KCNA1*, and

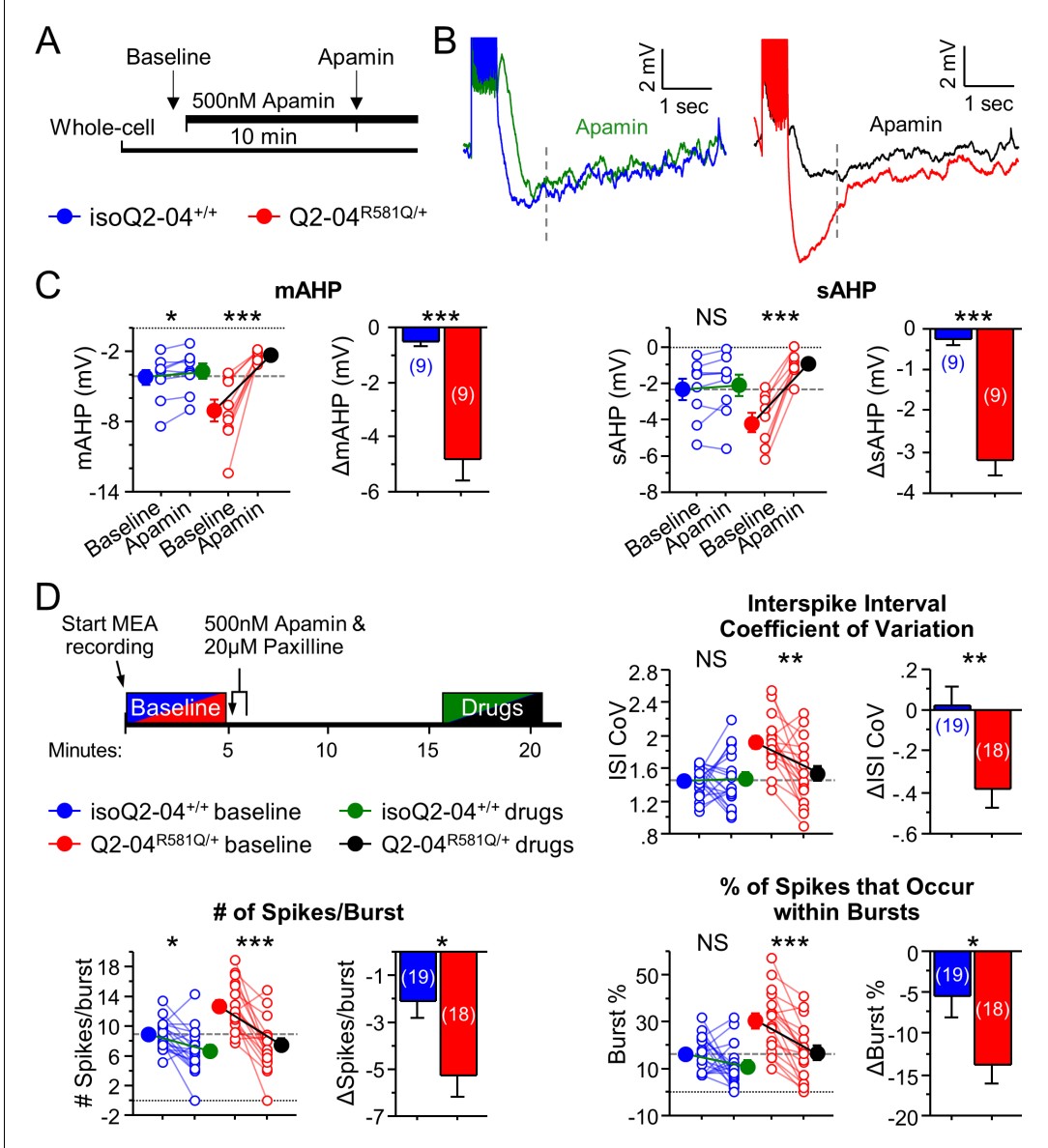

**Figure 5.** KCNQ2-DEE neurons exhibit a dyshomeostatic increase in SK and BK channel function. (**A**) Experimental protocol. Baseline measures were made after establishing the whole-cell configuration in current-clamp mode. After exactly 10 min of continuous perfusion of 500 nM apamin in aCSF, the AP properties and post-burst AHP were measured. (**B**) Representative traces showing post-burst AHPs before and after apamin application. (**C**) Acute application of apamin significantly reduced mAHP (***p=0.0004) and sAHP (***p<0.0001) in Q2-04^R581Q/+ neurons and reduced mAHP in isogenic control neurons (mAHP: *p=0.0114) but not sAHP (sAHP p=0.132). However, the magnitude by which apamin reduced both mAHP and sAHP was significantly larger in Q2-04^R581Q/+ neurons (t-test: ***p=0.0001, ***p<0.0001, respectively). (**D**) Right: Experimental protocol. Apamin (500 nM) and paxilline (20 μM) were added to MEAs after 5 min of baseline spontaneous recordings. The effect of drugs was measured after 10 min of continuous recording for 5 min. These K⁺ channel inhibitors reduced ISI CoV (**p=0.0015), spikes/burst (***p<0.0001) and % of all spikes that occur in bursts (***p<0.0001) in Q2-04^R581Q/+ neurons. The number of spikes per burst were reduced in isoQ2-04^+/+ neurons (*p=0.0051) but ISI CoV and burst % were not changed (p=0.796, p=0.0687, respectively). The magnitude by which apamin and paxilline reduced ISI CoV, spikes/burst and burst % was significantly larger in Q2-04^R581Q/+ neurons (t-test: **p=0.0048, *p=0.0087 and *p=0.0254, respectively). Repeated measures ANOVA was used to compare drug effects and *posthoc* t-tests were used where the interaction of before/after drug and genotype was significant. A dotted line is drawn through the mean baseline values measured for isoQ2-04^+/+ neurons before acute application of apamin and paxilline. Number of neurons analyzed is displayed within the figure; values displayed are mean ± SEM.

The online version of this article includes the following source data for figure 5:

**Source data 1.** Quantification of current-clamp parameters before and after acute apamin treatment for *Figure 5A–C*.

**Source data 2.** Quantification of MEA data parameters before and after acute apamin and paxilline treatment for *Figure 5D* and *Figure 6—figure supplement 3*.

*HCN1* was significantly higher in chronically XE991-treated isogenic control neurons compared to untreated neurons (p=0.0038, p=0.0148, p=0.0179, p=0.0307, p=0.0175, p=0.0443, and p=0.0002, respectively; *Figure 6G* and *Figure 6—figure supplement 1E*). Expression of other channel genes including *KCND2, KCNT1, KCNA2, KCNA4, KCNQ2, KCNQ3, KCNQ5,* and *HCN2* was not different following chronic XE991 treatment (*Figure 6—figure supplement 1E*).

Lastly, acute application of paxilline and apamin on MEAs during week 5, restored the bursting behavior of Q2-04$^{R581Q/+}$ neurons and chronically XE991-treated isogenic control neurons to the levels of untreated control neurons (*Figure 5* and *Figure 6—figure supplement 3A,B*). Collectively, these data suggest that upregulation of SK and BK channel function exacerbates bursting in Q2-04$^{R581Q/+}$ neurons, and is a result of reduced M-current.

## Discussion

We developed and studied a patient-specific iPSC-based model of KCNQ2-DEE that provided novel insight into the pathogenic mechanisms evoked by dysfunction of this ion channel. Our study demonstrated that neurons derived from an iPSC line heterozygous for a loss-of-function *KCNQ2* mutation exhibited progressive escalation of burst firing and developed intrinsic membrane properties that promoted phasic bursting as they matured over time in culture. This altered pattern of neuronal firing featured properties not previously associated with loss of M-current (faster AP repolarization and enhanced AHP). Our findings suggest that KCNQ2 dysfunction induces dyshomeostatic plasticity and alters the neurodevelopmental trajectory of KCNQ2-DEE neurons.

Epilepsy is a chronic condition with recurrent, paroxysmal, unprovoked seizures associated with specific EEG patterns. Repeated firing in bursts of high-frequency action potentials has been associated with chronic epilepsy both in experimental models and in humans (*Sanabria et al., 2001*; *Schindler et al., 2006*; *Gast et al., 2016*). Our analysis of firing patterns using MEAs indicated that KCNQ2-DEE neurons became active earlier and were more prone to fire in bursts rather than single tonic spikes exhibited by isogenic control neurons. As KCNQ2-DEE neurons matured, their bursts became increasingly pronounced with more spikes/burst and more phasic or regularly distributed intervals. This is supported by the combination of increased post-burst AHP and the later onset of enhanced AP repolarization in single neurons, which results in increased number of spikes per burst and longer refractory periods between bursts. This burst-suppression firing pattern is reminiscent of the interictal EEG pattern observed in KCNQ2-DEE patients (*Steriade, 2004*; *Timofeev and Steriade, 2004*; *Weckhuysen et al., 2012*; *Millichap et al., 2016*). Importantly, this type of activity does not necessarily reflect more neuronal APs, but rather an alteration in the neuronal discharge pattern (i.e. bursts rather than irregular single spikes) and propensity of KCNQ2-DEE neurons to fire within bursts.

Neurons dynamically adjust the expression and function of ion channels as well as the structure of their processes to regulate intrinsic excitability in response to cell autonomous defects or the environment. For example, during chronically induced hyperexcitability, neurons downscale their intrinsic excitability and alter the size and location of the AIS (*Turrigiano and Nelson, 2004*; *Grubb and Burrone, 2010*; *Wolfart and Laker, 2015*). Interestingly, Biba et al., recently reported that pyramidal neurons in heterozygous knock-in mice harboring the loss-of-function pathogenic T274M variant, exhibited hyperexcitability early on (P7-P9), but this effect went away in later life (P28-35; *Biba et al., 2020*). While the involvement of other channels, or the AHP specifically, were not examined, these results suggest that homeostatic changes also take place in neurons in vivo as a consequence of mutant *KCNQ2* . Furthermore, Lezmy et al. reported that M-current inhibition with XE991 acutely increased the firing rate of cultured hippocampal pyramidal neurons but chronic XE991 treatment restored firing to baseline levels. By contrast in cultured GABAergic interneurons, XE991 treatment caused a persistent hyperexcitability that was not attenuated over time (*Lezmy et al., 2020*). This suggests that homeostatic adaptive responses may be cell type specific. Importantly, a recent study showed that selective deletion of M-channels (Kv7.2/Kv7.3) in parvalbumin GABAergic interneurons leads to their hyperexcitability and enhanced excitatory transmission in pyramidal neurons (*Soh et al., 2018*). As GABA has been shown to have excitatory effects in early developing neurons it is possible that this enhanced excitatory transmission was due to excitatory GABA (*Cherubini et al., 1991*). An inherent limitation of our model system is that it is exclusively comprised of glutamatergic excitatory neurons. While this homogeneity infers a methodological advantage, the

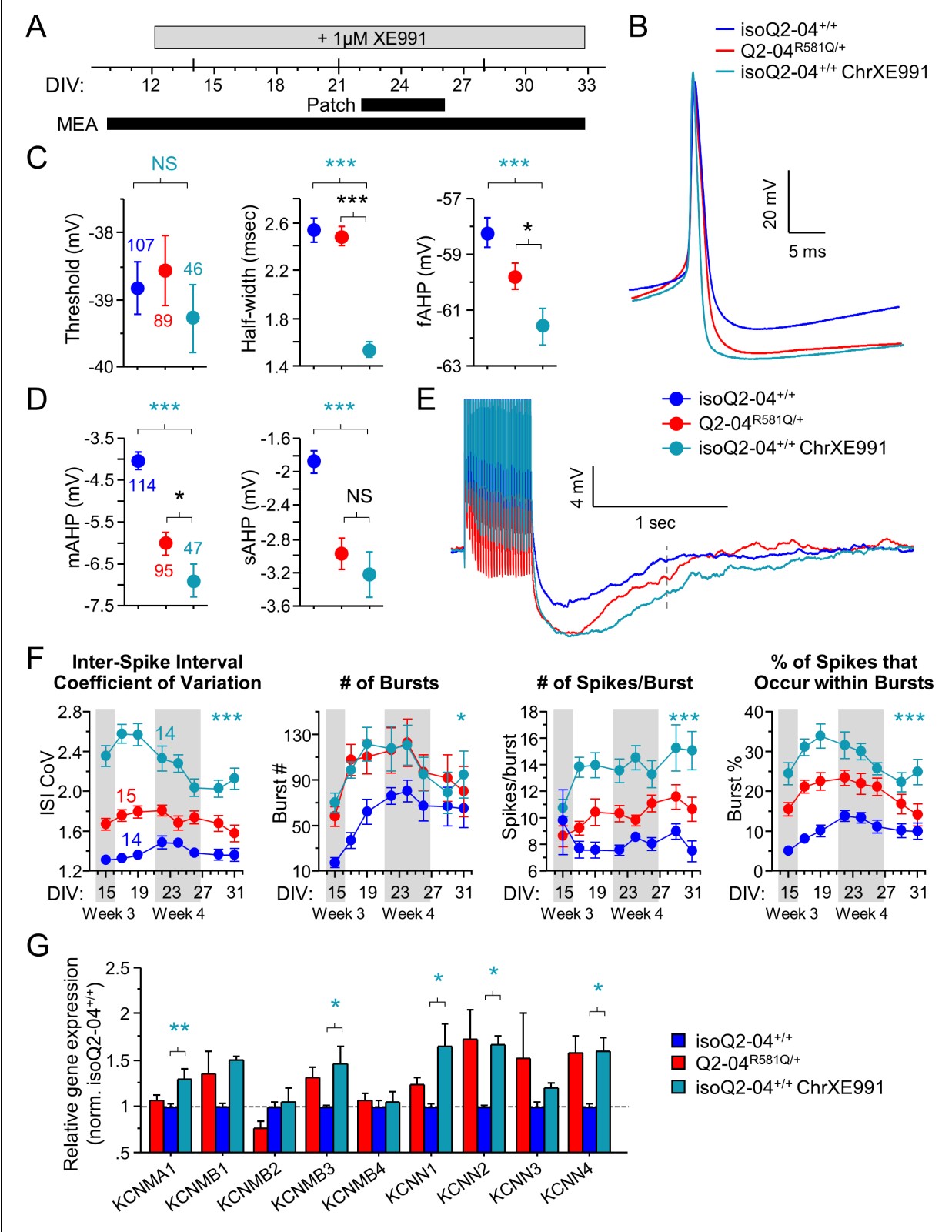

**Figure 6.** Chronic inhibition of M-current in control neurons phenocopies KCNQ2-DEE. (**A**) Experimental time line. (**B**) Representative AP traces from Q2-04$^{R581Q/+}$, untreated isoQ2-04$^{+/+}$ and isoQ2-04$^{+/+}$ neurons chronically treated with 1 μM XE991 (isoQ2-04$^{+/+}$ChrXE991). (**C**) Chronic XE991 treatment did not change the AP threshold but AP half-width and fAHP of isoQ2-04$^{+/+}$ChrXE991 were significantly different from untreated isogenic control neurons (ANOVA, Fisher's PLSD *posthoc* test for AP half-width: ***p<0.0001; fAHP: ***p=0.0001), and also from Q2-04$^{R581Q/+}$ patient-derived neurons

*Figure 6 continued on next page*

Figure 6 continued

(ANOVA, Fisher's PLSD posthoc test for AP half-width: p<0.0001; fAHP: *p=0.047). (D) The post-burst AHP is enhanced after chronic XE991 treatment in isoQ2-04$^{+/+}$ neurons (ANOVA, Fisher's PLSD posthoc test for mAHP: ***p<0.0001; sAHP: ***p<0.0001), to levels slightly larger or similar to Q2-04$^{R581Q/+}$ patient-derived neurons (mAHP: *p=0.038; sAHP: p=0.405). (E) Representative traces showing post-burst AHPs of Q2-04$^{R581Q/+}$, untreated isoQ2-04$^{+/+}$ and chronically XE991-treated isoQ2-04$^{+/+}$ neurons. Number of neurons analyzed is displayed within the figure and in **Tables 1** and **2** (also see **Figure 6—figure supplement 1**). NS: not significant. (F) MEA recordings from days 15 to 31 in culture recorded from Q2-04$^{R581Q/+}$, untreated isoQ2-04$^{+/+}$ and chronically XE991-treated isoQ2-04$^{+/+}$ neurons. Chronic XE991 treatment increased the ISI CoV, number of bursts, number of spikes/burst and burst % in chronically XE991-treated isoQ2-04$^{+/+}$ neurons compared to untreated isoQ2-04$^{+/+}$ (repeated measures ANOVA, Fisher's PLSD posthoc test: ***p<0.0001; *p=0.0275; ***p<0.0001; ***p<0.0001, respectively). Teal * indicate difference between chronically XE991-treated isoQ2-04$^{+/+}$ and untreated isoQ2-04$^{+/+}$ neurons. IsoQ2-04$^{+/+}$ chrXE991 neurons exhibited a similar or significantly greater bursting phenotype compared to untreated Q2-04$^{R581Q/+}$ neurons (repeated measures ANOVA, Fisher's PLSD *posthoc* test for ISI CoV: ***p<0.0001; number of bursts: p=0.9208; number of spikes/burst: ***p<0.0001; burst %: **p=0.0005). Fourteen to 15 wells were analyzed per group from two independent differentiations (see **Figure 6—figure supplements 1** and **2**). Repeated measures ANOVA was used to compare the three groups of neurons over the time course and Fisher's PLSD posthoc test was used only where there was significance between the groups. (G) Comparison of qPCR gene expression pattern of K$^+$ channels and major β subunits involved in AP repolarization and post-burst AHP among Q2-04$^{R581Q/+}$, untreated isoQ2-04$^{+/+}$ and chronically XE991-treated isoQ2-04$^{+/+}$ neurons at week 5. Expression of *KCNMA1, KCNMB3, KCNN1, KCNN2* and *KCNN4* was significantly higher in chronically XE991-treated isoQ2-04$^{+/+}$ neurons compared to untreated isoQ2-04$^{+/+}$ (ANOVA, Fisher's PLSD posthoc test: **p=0.0038, *p=0.0148, *p=0.0179, *p=0.0307 and *p=0.0175, respectively). Values displayed are from two independent differentiations normalized within each differentiation to isoQ2-04$^{+/+}$. Teal * and $^\#$ indicate significance between chronically XE991-treated and untreated isoQ2-04$^{+/+}$ neurons. Values displayed are mean ± SEM.

The online version of this article includes the following source data and figure supplement(s) for figure 6:

**Source data 1.** Quantification of passive and active current-clamp parameters with chronic XE991 treatment for **Figure 6A–E** and **Figure 6—figure supplement 1A and B**.

**Source data 2.** Quantification of DIV 15–31 MEA data parameters with chronic XE991 treatment for **Figure 6F** and **Figure 6—figure supplement 1C and D**.

**Source data 3.** Quantification of RT-qPCR expression ΔΔCt values for **Figure 6G** and **Figure 6—figure supplement 1E**.

**Figure supplement 1.** Intrinsic membrane properties and MEA recordings in chronically XE991-treated control neurons.

**Figure supplement 2.** Early effects of chronic XE991 treatment on control neurons.

**Figure supplement 2—source data 1.** Quantification of DIV 10–13 MEA data parameters for **Figure 6—figure supplement 2**.

**Figure supplement 3.** Effects of acute paxilline/apamin treatment on chronically XE991-treated control neurons.

**Figure supplement 4.** Diagram of proposed temporal homeostatic consequences of loss of M-current.

integration of GABAergic inhibitory neurons, which are critical in the synchronization of firing during epileptic activity, will provide a more physiological iPSC-based model for interrogating the effects of *KCNQ2* mutations.

Homeostatic plasticity can become maladaptive and even pathogenic, when these processes become dysfunctional (*Moulder et al., 2003*; *Turrigiano and Nelson, 2004*; *Wu et al., 2008*; *Miranda et al., 2013*; *Swann and Rho, 2014*; *Wolfart and Laker, 2015*; *Wefelmeyer et al., 2016*). On the other hand, the 'acquired channelopathy' hypothesis suggests that proepileptic ion channel dysfunction develops during or after the onset of epilepsy or excitotoxicity (*Bernard et al., 2004*; *Poolos and Johnston, 2012*). For example, fast upregulation of BK channels and associated potassium currents occurs in cortical neurons of mice after picrotoxin-induced seizure pre-sensitization (*Shruti et al., 2008*). However, these mechanisms are not mutually exclusive and it is plausible that episodic hyperexcitability can promote both protective homeostasis and epileptogenic neuronal adaptation.

Our findings are concordant with previous work describing homeostatic responses to chronic M-current inhibition and provide further insight into the mechanisms that mediate this adaptive response by the upregulation of other ion conductances including BK and SK channels (*Okada et al., 2003*; *Lezmy et al., 2017*; *Lezmy et al., 2020*). We demonstrated that suppression of M-current leads to rearrangement of neuronal intrinsic properties on different temporal scales (*Figure 6—figure supplement 4*). Inhibition of M-current by application of XE991 increases the number of spontaneously active neurons and their ability to fire in bursts within 24 hr (*Figure 6—figure supplement 2*). Interestingly, the magnitude of change in firing frequency of control neurons after a 24 hr treatment with XE991 was not significantly different from untreated neurons, while the change in % of spikes that occur within bursts was dramatically enhanced. Thus, our finding that burst firing is progressively and dramatically increased in KCNQ2-DEE neurons signifies a necessity to shift how

**Table 2.** Intrinsic membrane properties of week 4 isogenic control neurons chronically treated with XE991.

|  | isoQ2-04[+/+] chronic XE991 week 4 |
| --- | --- |
| Resting potential (mV) | −57.4 ± 1 (54) ** |
| Input resistance $R_N$ at RMP (MΩ) | 644 ± 32.1 *** |
| Series resistance $R_S$ RMP (MΩ) | 11.3 ± 0.3 ** |
| AP amplitude from baseline (mV) | 93.1 ± 1 (46) |
| AP threshold (mV) | −39.3 ± 0.5 |
| AP half-width (ms) | 1.5 ± 0.1 *** |
| fAHP (mV) | −61.6 ± 0.7 * |
| mAHP (mV) | −6.9 ± 0.4 (47) *** |
| sAHP (mV) | −3.2 ± 0.3 *** |

*$p < 0.05$, **$p < 0.005$, ***$p < 0.0005$: posthoc Fisher PLSD test comparing untreated isoQ2-04[+/+] neurons at week 4 (see **Table 1**) with chronically XE991-treated isoQ2-04[+/+] neurons recorded on week 4. Number of neurons is indicated in (). RMP - resting membrane potential; AP - Action potential; AHP - afterhyperpolarization; fAHP - fast AHP; mAHP - medium AHP; sAHP - slow post-burst AHP.

we examine neuronal disease phenotypes and focus on the pattern rather than abundance of activity.

A greater number of active neurons during early time points could be directly related to the loss of M-current, which induces fast homeostatic rearrangement of intrinsic properties to upregulate post-burst AHPs and limit neuronal firing, as well as much slower dyshomeostatic enhancement of AP repolarization (by weeks 4 and 5). KCNQ2-DEE neurons, examined at week 3, exhibited some properties consistent with a loss of M-current (i.e. hyperpolarized RMP, increased IR, longer AP half-widths and a trend for smaller fAHPs; *Figure 3* and *Figure 3—figure supplement 1A,B*) and associated with a bursting phenotype. However, soon after (week 4 and beyond) KCNQ2-DEE neurons developed shorter AP half-widths and larger fAHPs associated with increased spikes/burst, as well as regularity of burst firing. This suggests a maladaptive upregulation of fast voltage-gated K$^+$ conductances over time. Accordingly, the upregulation in *KCNMB3* expression in KCNQ2-DEE neurons partially explains the increased AP repolarization as β3 subunits transform BK channels to be fast-inactivating (*Storm, 1989*; *Xia et al., 1999*; *Hu et al., 2003*; *Zeng et al., 2007*; *Kaufmann et al., 2010*; *Li and Yan, 2016*; *Latorre et al., 2017*). The upregulation of *KCNA1* and *KCNA2* is also consistent with enhanced AP repolarization (*Colasante et al., 2020*). Chronic M-current inhibition in isoQ2-04[+/+] neurons enhanced the expression of *KCNMA1*, *KCNMB3* and *KCNA1* which could explain why AP repolarization in these neurons is even faster than in KCNQ2-DEE neurons.

Conversely, the magnitude of the mAHP and sAHP did not change over time in KCNQ2-DEE neurons, suggesting that either the driver for post-burst AHPs is upregulated early and fails to be developmentally downregulated (as the AHP in isoQ2-04[+/+] neurons becomes smaller at week 4; *Figure 3E,F*), or there are multiple mechanisms through which the AHP is enhanced. Activation of Ca$^{2+}$-activated K$^+$ channels underlies the medium and slow AHP (*Lancaster and Adams, 1986*; *Zhang and McBain, 1995*). We observed enhanced, apamin-sensitive upregulation of post-burst AHPs in KCNQ2-DEE neurons, which correlated with increased *KCNN1* and *KCNN2* gene expression at week 5. The magnitude of the post-burst AHP is positively related to the rise of intracellular Ca$^{2+}$ that accumulates with repeated AP firing. Higher frequency and longer duration of activity leads to greater influx and accumulation of intracellular Ca$^{2+}$ (for review, see *Storm, 1990*). As KCNQ2-DEE neurons exhibit more spikes/burst, it is also likely that a combination of increased SK channel expression and activity-dependent Ca$^{2+}$ accumulation contribute to the enhanced AHP in KCNQ2-DEE neurons. Furthermore, higher expression of *KCNMA1* and *KCNN4* (encoding an apamin-insensitive IK channel) at week 3 but not week 5 suggests multiple developmentally regulated molecular mechanisms that drive post-burst AHPs. BK channel α-subunit alone does not inactivate and thus can participate in the post-burst AHP (*Storm, 1989*; *Xia et al., 1999*; *Kaufmann et al., 2010*; *Li and Yan, 2016*; *Latorre et al., 2017*). Moreover, chronic XE991 treatment increased expression of *KCNMA1*,

*KCNN1, KCNN2*, and *KCNN4* at week 5 suggesting that stronger suppression of M-current may exacerbate dyshomeostatic upregulation of these channels. Interestingly, SK2 (*KCNN2*) overexpression or activation impairs cognitive function and learning and memory (*Hammond et al., 2006*; *McKay et al., 2012*).

Our experiments with apamin and paxilline suggest that both SK and BK channels contribute to the increased K$^+$ conductance that enhances AP repolarization as well as AHPs, and leads to phasic bursting in KCNQ2-DEE neurons. Inhibition of these channels with apamin and paxilline restored bursting behavior of KCNQ2-DEE and chronically XE991-treated isoQ2-04$^{+/+}$ neurons (*Figure 5D* and *Figure 6—figure supplement 3*). Although, increased K$^+$ conductance may not be traditionally associated with hyperexcitability and epilepsy, several studies have shown that inhibition of K$^+$ conductances improves learning and memory in wild-type mice, or seizure susceptibility in a model of Angelman syndrome (*Stackman et al., 2002*; *Fontán-Lozano et al., 2011*; *Sun et al., 2019*). Furthermore, gain-of-function mutations in several K$^+$ channels including KCNQ2, KCNQ3, KCNQ5, BK, SK3, KCNT1, and K$_V$4.2 have been reported in genetic epilepsy and developmental disorders (*Niday and Tzingounis, 2018*; *Bauer et al., 2019*). Because KCNQ2-DEE patients present with seizures in the first days of life, targeting the underlying cause might not be effective if started late in the disease course. Our study suggests that targeting dyshomeostatically altered ion currents might offer an alternative therapeutic strategy for the cognitive and developmental deficits in KCNQ2-DEE. The iPSC-based platform we developed may be valuable to identify effective therapeutics and address further questions regarding the spatiotemporal mechanisms of DEE due to *KCNQ2* mutations.

# Materials and methods

## Key resources table

| Reagent type (species) or resource | Designation | Source or reference | Identifiers | Additional information |
|---|---|---|---|---|
| Antibody | Anti-GFP (Chicken polyclonal) | Abcam | Cat# ab13970, RRID: AB_300798 | ICC (1:10,000) |
| Antibody | Anti-Nanog (Goat polyclonal) | R and D Systems | Cat# AF1997, RRID:AB_355097 | ICC (1:500) |
| Antibody | Anti-hSSEA4 (Mouse monoclonal) | DSHB | Cat# MC-813–70, RRID:AB_528477 | ICC (3 µg/ml) |
| Antibody | Anti-MAP2 (Mouse monoclonal) | Millipore | Cat# MAB3418, RRID: AB_94856 | ICC (1:1000) |
| Antibody | Anti-vGLUT1 (Rabbit polyclonal) | Synaptic Systems | Cat# 135 302, RRID:AB_887877 | ICC (1:200) |
| Antibody | Anti-chicken secondary Alexa Fluor 488 (Goat polyclonal) | Thermo Fisher Scientific | Cat# A-11039, RRID:AB_142924 | ICC (1:1000) |
| Antibody | Anti-mouse secondary Alexa Fluor 568 (Goat polyclonal) | Thermo Fisher Scientific | Cat# A-11031, RRID:AB_144696 | ICC (1:1000) |
| Antibody | Anti-rabbit secondary Alexa Fluor 647 (Goat polyclonal) | Thermo Fisher Scientific | Cat# A-21245, RRID:AB_141775 | ICC (1:1000) |
| Antibody | Anti-goat secondary Alexa Fluor 568 (Donkey polyclonal) | Thermo Fisher Scientific | Cat# A-11031, RRID:AB_144696 | ICC (1:500) |
| Antibody | Anti-mouse secondary Alexa Fluor 488 (Donkey polyclonal) | Thermo Fisher Scientific | Cat# A-21202, RRID:AB_141607 | ICC (1:500) |
| Chemical Compound, Drug | XE991 | Abcam | Cat# ab120089 | |
| Chemical Compound, Drug | XE991 | TOCRIS | Cat# 2000 | |
| Chemical Compound, Drug | Ara-C | Sigma-Aldrich | Cat# C1768 | |

*Continued on next page*

Continued

| Reagent type (species) or resource | Designation | Source or reference | Identifiers | Additional information |
|---|---|---|---|---|
| Chemical Compound, Drug | TRIzol | Invitrogen | Cat# 15596026 | |
| Chemical Compound, Drug | ProLong Gold Antifade Mountant | Thermo Fisher Scientific | RRID:SCR_015961 | |
| Chemical Compound, Drug | Poly-d-Lysine | Sigma-Aldrich | Cat# P6407-5MG | |
| Chemical Compound, Drug | Laminin | Thermo Fisher Scientific | Cat# 23017–015 | |
| Chemical Compound, Drug | LDN-193189 | DNSK International | Cat# 1062368-24-4 | |
| Chemical Compound, Drug | SB431542 | DNSK International | Cat# 301836-41-9 | |
| Chemical Compound, Drug | XAV939 | DNSK International | Cat# 384028-89-3 | |
| Chemical Compound, Drug | Doxycycline | Sigma-Aldrich | Cat# D9891-5G | |
| Chemical Compound, Drug | Puromycin | Sigma-Aldrich | Cat# P8833 | |
| Chemical Compound, Drug | BDNF | R and D Systems | Cat# 248-BD | |
| Chemical Compound, Drug | Heparin Sulfate | Sigma-Aldrich | Cat# H3149 | |
| Chemical Compound, Drug | Knockout DMEM | Thermo Fisher Scientific | Cat# 10829–018 | |
| Chemical Compound, Drug | Knockout Replacement Serum | Thermo Fisher Scientific | Cat# 10828–010 | |
| Chemical Compound, Drug | MEM non-essential amino acids | Thermo Fisher Scientific | Cat# 11140–076 | |
| Chemical Compound, Drug | Glutamax | Thermo Fisher Scientific | Cat# 35050–061 | |
| Chemical Compound, Drug | 2-Mercaptoethanol | Thermo Fisher Scientific | Cat# 21985–023 | |
| Chemical Compound, Drug | DMEM/F12 + L-glutamine | Thermo Fisher Scientific | Cat# 11320–082 | |
| Chemical Compound, Drug | 45% glucose solution | Sigma-Aldrich | Cat# G8769-100ML | |
| Chemical Compound, Drug | N2 supplement | Thermo Fisher Scientific | Cat# 17502–048 | |
| Chemical Compound, Drug | B27 supplement | Thermo Fisher Scientific | Cat# 17504–044 | |
| Chemical Compound, Drug | Neurobasal + L-glutamine | Thermo Fisher Scientific | Cat# 21103–049 | |
| Chemical Compound, Drug | MEM | Thermo Fisher Scientific | Cat# 10370–021 | |
| Chemical Compound, Drug | Horse Serum | Thermo Fisher Scientific | Cat# 26050–070 | |
| Chemical Compound, Drug | Hyclone FBS | VWR | Cat# 16777–006 | |
| Chemical Compound, Drug | DNase I, Amplification Grade | Life Technologies | Cat# 18-068-015 | |
| Chemical Compound, Drug | DNase | Worthington Biochemical Corp. | Cat# LK003172 | |

*Continued*

| Reagent type (species) or resource | Designation | Source or reference | Identifiers | Additional information |
|---|---|---|---|---|
| Chemical Compound, Drug | Paxilline | Alomone Labs | Cat# P-450 | |
| Chemical Compound, Drug | Apamin | Alomone Labs | Cat# STA-200 | |
| Commercial Assay, Kit | Invitrogen CytoTune - iPS 2.0 Sendai reprogramming kit | Life Technologies | Cat# A16517 | |
| Commercial Assay, Kit | Wizard SV Gel and PCR Clean-Up System | Promega | Cat# A9281 | |
| Commercial Assay, Kit | DNeasy Blood and Tissue Kit | Qiagen | Cat# 69504 | |
| Commercial Assay, Kit | MycoAlert PLUS Detection Kit | Lonza | Cat# LT07-710 | |
| Commercial Assay, Kit | MycoAlert Assay Control Set | Lonza | Cat# LT07-518 | |
| Commercial Assay, Kit | Invitrogen SuperScript IV First-Strand Synthesis System | Thermo Fisher Scientific | Cat# 18-091-050 | |
| Commercial Assay, Kit | Genecopoeia T7 Endonuclease I Kit | Genecopoeia | Cat# IC005 | |
| Commercial Assay, Kit | QuikChange II XL Site-Directed Mutagenesis Kit | Agilent technologies | Cat# 200522 | |
| Commercial Assay, Kit | iTaq Universal SYBR Green Supermix | Bio-Rad | Cat# 1725124 | |
| Commercial Assay, Kit | Pooled human fetal brain total RNA | TakaRa Bio | #636526, Lot #1612396A | |
| Commercial Assay, Kit | Pooled adult cortex total RNA | TakaRa Bio | #636561, Lot #2007106 | |
| Cell Line (*Homo-sapiens*) | Human: KCNQ2-DEE and mutation corrected isogenic patient-derived iPSCs | This paper | N/A | See Materials and methods |
| Cell Line (*Homo-sapiens*) | Human: 11a | Harvard University; Boston; USA | RRID: CVCL_8987 | (Control 1) |
| Cell Line (*Homo-sapiens*) | Human: NCRM-5 | NIH Center for Regenerative Medicine – Bethesda | RRID: CVCL_1E75 | (Control 2) |
| Cell Line (*Homo-sapiens*) | Human: 18a | Harvard University; Boston; USA | RRID:CVCL_8993 | (Control 3) |
| Cell Line (*M. musculus*) | Primary culture (CD-1 IGS) Mouse Glia | Charles River | | See Materials and methods |
| Cell Line (*C. griseus*) | KCNQ3 stable CHO cells | American Type Culture Collection; This paper | ATCC Cat# CRL-9618, RRID:CVCL_0214 | Stable expression of KCNQ3 (See Materials and methods) |
| Recombinant DNA Reagent | pCS2_KCNQ2 _IRES2_EGFP | This paper | | See Materials and methods |
| Recombinant DNA Reagent | pcDNA5/FRT_KCNQ3 | This paper | | See Materials and methods |
| Recombinant DNA Reagent | pTet-O-Ngn2-puro | Addgene | RRID:Addgene_52047 | |
| Recombinant DNA Reagent | Tet-O-FUW-EGFP | Addgene | RRID:Addgene_30130 | |
| Recombinant DNA Reagent | FUW-M2rtTA | Addgene | RRID:Addgene_20342 | |
| Sequence-based reagent | All qPCR and sequencing primer sequences | This paper | | See **Supplementary files 2** and 4 |

*Continued on next page*

*Continued*

| Reagent type (species) or resource | Designation | Source or reference | Identifiers | Additional information |
|---|---|---|---|---|
| Sequence-based reagent | R581Q mutagenesis F | This paper | PCR primers | TCCC**A**AATTAAGAGCCTGC AGTCCAGAGTGGAC |
| Sequence-based reagent | R581Q mutagenesis R | This paper | PCR primers | AGGCTCTTAATT**T**GGGACA GCATGTCCAGGTGGC |
| Software, Algorithm | Fiji (ImageJ) | Max Planck Institute | RRID:SCR_002285 | |
| Software, Algorithm | Prism 5.0 | GraphPad | RRID:SCR_002798 | |
| Software, Algorithm | NIS-Elements | Nikon | RRID:SCR_002776 | |
| Software, Algorithm | Image Lab | Bio-Rad | RRID:SCR_014210 | |
| Software, Algorithm | Metamorph | Molecular Devices | RRID:SCR_002368 | |
| Software, Algorithm | AxIS | Axion biosystems | RRID:SCR_016308 | |
| Software, Algorithm | Neural Metrics Tool: Neural Module | Axion biosystems | RRID:SCR_019270 | |
| Software, Algorithm | AxIS Metric Plotting Tool | Axion biosystems | RRID:SCR_016308 | |
| Software, Algorithm | Pclamp/ Clampfit | Molecular Devices | RRID:SCR_011323 | |
| Software, Algorithm | MATLAB | MATLAB mathworks | RRID:SCR_001622 | |
| Software, Algorithm | Statview 5.0 | SAS Institute Inc | RRID:SCR_017411 | |
| Software, Algorithm | Snapgene | Snapgene | RRID:SCR_015052 | |
| Software, Algorithm | CFX Manager | Bio-Rad | RRID:SCR_017251 | |
| Software, Algorithm | FASTQC | Braham Institute | RRID:SCR_014583 | |
| Software, Algorithm | BWA | Wellcome Trust Sanger Institute | RRID:SCR_010910 | |
| Software, Algorithm | GATK | Broad Institute | RRID:SCR_001876 | |
| Software, Algorithm | VCFTools | 1000 Genomes Project Analysis Group | RRID:SCR_001235 | |
| Software, Algorithm | PLINK | cog-genomics/ Broad Institute | RRID:SCR_001757 | |
| Software, Algorithm | Circos | Michael Smith Genome Sciences Centre | RRID:SCR_011798 | |
| Software, Algorithm | CADD | University of Washington | RRID:SCR_018393 | Database |
| Software, Algorithm | gnomAD | Broad Institute | RRID:SCR_014964 | Database |

## Cell lines

Control 1 and 3 hiPSC line (11a and 18a; RRID:CVCL_8987 and RRID:CVCL_8993) was derived previously (*Boulting et al., 2011*). Control two hiPSC line (NCRM-5; NHCDR Cat# ND50031, RRID:CVCL_1E75) was obtained from the NIH Center for Regenerative Medicine (NIH CRM). KCNQ2-DEE patient-derived (Q2-04$^{R581Q/+}$) and isogenic control (isoQ2-04$^{+/+}$) iPSC lines were derived as described below. Further information on all iPSC lines can be found in *Figure 1—figure supplement 4B*.

## Preparation of plasmids and lentivirus

Full-length cDNA encoding WT human KCNQ2 splice isoform 4 cDNA (K$_V$7.2; GenBank accession NM_172108) was engineered in the mammalian expression vector pIRES2_EGFP or a modified vector where EGFP was substituted by CyOFP1. Site-directed mutagenesis of KCNQ2 was performed using QuikChange II XL (Agilent technologies, Santa Clara, CA, USA; mutagenic primer sequences: 5': TCCCAAATTAAGAGCCTGCAGTCCAGAGTGGAC, 3': AGGCTCTTAATT**T**GGGACAGCATG TCCAGGTGGC) to insert the R581Q (R550Q in isoform 4) patient mutation into the wildtype construct. KCNQ3 (K$_V$7.3; GenBank accession NM_004519) was cloned into pcDNA5/FRT for use in generating the KCNQ3-stable CHO-K1 cells.

TetO-Ngn2-puro (Addgene plasmid #52047) and TetO-FUW-EGFP (Addgene plasmid #30130) plasmids were gifts from Marius Wernig (*Vierbuchen et al., 2010*; *Zhang et al., 2013*). FUW-M2rtTA (Addgene plasmid # 20342) was a gift from Rudolf Jaenisch (*Hockemeyer et al., 2008*). Lentiviruses were generated in HEK293T cells using the second-generation packaging vectors, psPAX2 and pMD2.G, as described previously (*Zufferey et al., 1998*) by the Northwestern University DNA/RNA Delivery Core.

## Generation of iPSCs

Peripheral blood mononuclear cells (PBMCs) were isolated from whole blood following informed consent under protocols approved both by Ann and Robert H. Lurie Children's Hospital of Chicago and Northwestern University. Reprogramming of PBMCs into iPSCs was performed at the Northwestern Stem Cell Core Facility using Invitrogen's CytoTune-iPS 2.0 Sendai Reprogramming system (A16517, Thermofisher), following the manufacturer's instructions. This reprogramming system uses four transcription factors (Oct4, Sox2, Klf4, c-Myc). Briefly, PBMCs ($5 \times 10^5$) were seeded into one well of a 24-well plate and cultured for four days in StemSpan SFEM II PBMC complete medium (STEMCELL Technology, 09655) supplemented with 100 ng/ml SCF (PeproTech, 300–07), 100 ng/ml FLT3 (PeproTech, 300–19), 20 ng/ml IL-3 (PeproTech, 200–03), and 20 ng/ml IL-6 (PeproTech, 200–06). Immediately after plating, the cells were infected with Sendai virus for 48 hr at 37°C. The infected cells were transferred onto MEF feeders and cultured in StemSpan SFEM II. Following 21–28 days of culture, individual iPSC colonies were picked and transferred to Matrigel (BD Biosciences, BD354277) coated six-well plate for expansion and were maintained in mTeSR1 (STEMCELL Technology, 85850).

## Cell culture

Chinese hamster ovary cells (CHO-K1, CRL 9618, American Type Culture Collection, Manassas VA, USA) stably expressing human KCNQ3 were grown in F-12 nutrient mixture medium (GIBCO/Invitrogen, San Diego, CA, USA) supplemented with 10% fetal bovine serum (ATLANTA Biologicals, Norcross, GA, USA), Zeocin (100 µg/ml) and hygromycin B (600 µg/ml), penicillin (50 units·ml$^{-1}$), streptomycin (50 µg·ml$^{-1}$) at 37°C in 5% $CO_2$. Unless otherwise stated, all tissue culture media was obtained from Life Technologies, Inc (Grand Island, NY, USA). Plasmids encoding WT and/or variant KCNQ2 were transiently transfected into stable KCNQ3 cells by electroporation using the Maxcyte STX system (MaxCyte Inc, Gaithersburg, MD, USA) as reported previously (*Vanoye et al., 2018*). We studied the KCNQ2 R581Q variant in both the homozygous and heterozygous (1:1 with wild-type KCNQ2) state in cells stably expressing KCNQ3 subunits. For the homozygous channel experiments, 15 µg KCNQ2 wild-type or variant KCNQ2 DNA were electroporated. For the heterozygous channel experiments, 20 µg of KCNQ2 wild-type or 10 µg wild-type DNA plus 10 µg variant DNA were co-electroporated.

All iPSCs were grown on Matrigel with mTeSR1 media and passaged weekly using Accutase (Sigma). All cell cultures were maintained at 37°C and 5% $CO_2$. All cell lines were regularly tested for presence of mycoplasma using MycoAlert PLUS Detection Kit (Lonza) and determined to be mycoplasma-free.

Primary glial cell cultures were derived from postnatal day 0–2, CD-1 mice (Charles River). Briefly, brain cortices were dissected free of meninges in dissection buffer HBSS (Thermo Fisher), then digested with trypsin (Thermo Fisher) and DNAse I (Worthington) for 10 min at 37°C. The tissue was dissociated in glia medium: DMEM (Corning, #15–013-CV) supplemented with Glutamax, D-glucose, 10% normal horse serum (Life Technologies), and penicillin-streptomycin (Thermo Fisher). After centrifugation and resuspension, cells were filtered through a 0.45 micron cell strainer and plated on poly-D-lysine coated plates with glia media at 37°C, 5% $CO_2$ for 2 weeks. Afterwards, glial cultures were tested for mycoplasma, dissociated for expansion, and frozen in 10% DMSO/horse serum. All animal experiments were approved and conducted in accordance with the policies and guidelines set forth by the Northwestern University Institutional Animal Care and Use Committee (IACUC).

## CRISPR/Cas9 gene-editing

Isogenic control iPSCs were generated using CRISPR/Cas9 from the Q2-04$^{R581Q/+}$ patient-derived iPSC line in collaboration with Applied StemCell (Milpitas, CA). Briefly, one million patient iPSCs

were electroporated with a mixture of guide RNA (gRNA) and Cas9 (in the ribonucleoprotein format), and ssODN (*Supplementary file 1*). A small portion of the cell culture, presumably with mixed population, was subjected to Sanger sequencing analysis. Once the mixed culture showed repair with qualified HDR efficiency, the transfected cells were subjected to single cell cloning. Single-cell-derived clones were cultured for 15–20 days, followed by genotype analysis by Sanger sequencing. Clones with the desired genetic modification were identified by PCR genotyping and confirmed by Sanger sequencing (*Figure 1D*). All primers sequences can be found in *Supplementary file 2*.

## Analysis of off-target Cas9 sites and whole genome sequencing

Potential off-target sites were predicted with the online tool: CCTop CRISPR/Cas9 target online predictor https://cctop.cos.uni-heidelberg.de/ (*Stemmer et al., 2015*). We selected the top 10 genomic regions of homology and thus most likely off-target sites, and amplified each one by targeted PCR of genomic DNA from the corrected iPSC clone, for further analysis either by Sanger Sequencing or by a T7 Endonuclease assay (*Figure 1—figure supplement 3A*; *Supplementary files 1* and *2*). The same PCR conditions were used to amplify the positive control DNA template and primer mix, included in the T7 Endonuclease I Assay Kit (Genecopoeia). The amplified DNA from each potential off-target site was purified using the Wizard SV Gel and PCR Clean-Up System (Promega). The concentration of the purified DNA from the potential off-target sites and the template DNA from the positive control was assessed by using a Nanodrop 2000 Spectrophotometer (Thermo-Fisher). Amplicons (500 ng) from each potential off-target region of the isogenic and parental cell lines and 500 ng of the positive control DNA template were heated to 95°C for 5 min and subsequently allowed to cool to room temperature to denature and re-anneal the PCR products, respectively. T7 Endonuclease I was added to the re-annealed PCR products and incubated at 37°C for 60 min. The PCR products from the potential off-target sites and the positive control template were then run on a gel with 6x loading buffer, alongside a 2-log DNA ladder (New England BioLabs Inc). All primer sequences can be found in *Supplementary file 2*.

Whole genome sequencing was done on Q2-04$^{R581Q/+}$ and isoQ2-04$^{+/+}$ iPSC genomic DNA in collaboration with Novogene Corporation Inc The genomic DNA was randomly fragmented by sonication, then DNA fragments were end polished, A-tailed, and ligated with the full-length adapters for Illumina sequencing, and secondary PCR amplification with P5 and indexed P7 oligos. The PCR products used in the final construction of the libraries were purified with AMPure XP system. Libraries were checked for size distribution by Agilent 2100 Bioanalyzer (Agilent Technologies, CA, USA), and quantified by real-time PCR (to meet the criteria of 3 nM). Alignment to the human reference genome (build hg38) was done with the Burrows-Wheeler Aligner (BWA) v.0.7.17. Variation calling and quality controls were performed using the GATK best practices pipeline.

Whole genome sequencing reads were checked for quality using FASTQC (http://www.bioinformatics.babraham.ac.uk/projects/fastqc). Alignment to the human reference genome build hg38 was done using the Burrows-Wheeler Aligner (BWA v.0.7.17) (*Li and Durbin, 2009*). The variant calling process was performed following the Genome Analysis ToolKit *best practices* pipeline (GATK4; *McKenna et al., 2010*). Briefly, aligned reads in bam format followed duplicate marking, coordinate sorting and base quality score recalibration (BQSR). Next, Haplotype Caller was used to generate genomic variant call format (gVCFs) for each individual chromosome and GatherGVCF was used to concatenate them. Raw VCFs were generated using the genomicsDBimport and GenotypeGVCFs tools. Further QCs were performed using hard filtering and Variant Quality Score Recalibration (VQSR). Post-GATK QC's were done using VCFTools v.0.1.17 (*1000 Genomes Project Analysis Group et al., 2011*), removing variants without PASS in the FILTER field, with read depth (DP) <20 and genotype quality (GQ) <20. Indel left-normalization and multiallelic splitting were also performed before annotation.

Variant annotation and functional deleterious prediction were done using ANNOVAR (*Wang et al., 2010*). Potentially deleterious variants were selected if they had (i) frequency <0.01 (rare) in gnomAD v2.1.1 exome and genome cohorts *Genome Aggregation Database Consortium et al., 2020*; (ii) located in exonic and splicing sites; (iii) predicted to be nonsynonymous, frameshift and nonframeshift indels, and stop-gain and stop-loss, and (iv) CADD score of >12.37 top 2% of most deleterious variants; (*Rentzsch et al., 2019*). Whole-genome similarity comparison was performed by IBD analysis in PLINK v.1.9 (*Chang et al., 2015*). We used common variants (frequency >1%) and performed linkage-disequilibrium (LD) pruning. For single variant comparison, we

selected all non-missing positions with genotype differences (different zygosity) between samples Q2-04$^{R581Q/+}$ and isoQ2-04$^{+/+}$ (e.g. homozygous reference to heterozygous, homozygous reference to homozygous alternative, etc.). Variants with genotype differences were plotted as a circular ideogram using Circos plot (*Krzywinski et al., 2009*), generating densities in 1 MB windows, followed by potentially deleterious variants, genes harboring variants and CRISPR off-target sites.

## Cortical excitatory neuron differentiation

iPSCs were differentiated into cortical glutamatergic neurons using a modified version of a protocol based on *Ngn2* overexpression (*Zhang et al., 2013*). Stem cells were dissociated as single cells using Accutase, re-suspended in mTeSR1 with 10 µM ROCK inhibitor (Y-27632, DNSK International, #129830-38-2), then incubated with lentiviruses (FUW-M2rtTA, TetO-Ngn2-Puro, TetO-FUW-EGFP) in suspension for 5 min before plating (95,000 cells/cm$^2$; *Figure 1—figure supplement 4A*). After 24 hr (day 1), lentivirus was removed and replaced with knockout serum replacement medium (KOSR) consisting of KnockOut DMEM supplemented with Knockout replacement serum KSR, nonessential amino acids (NEAA), Glutamax (Life Technologies), 55 µM β-mercaptoethanol (Gibco, Cat# 21985023), 10 µM SB431542 (DNSK International), 100 nM LDN-193189 (DNSK International), 2 µM XAV939 (DNSK International), and 3 µg/ml of doxycycline (Sigma). On the following day (Day 2), media was replaced with a 1:1 ratio of KOSR to neural induction media (NIM) composed of DMEM: F12 supplemented with NEAA, Glutamax, N2 (Gibco, Life Technologies), 0.16% D-glucose (Sigma) and 2 µg/ml heparin sulfate (Sigma). Doxycycline (2 µg/ml) and puromycin (2 µg/ml; Sigma) were added to this NIM media. On Day 3, the media was replaced with NIM containing doxycycline (3 µg/ ml) and puromycin (2 µg/ml). All neurons were frozen in 10% DMSO/Hyclone FBS (VWR) on Day 4 (*Figure 1—figure supplement 4A*). For all experimental analysis, iPSC-derived neurons were plated on primary CD1 mouse cortical glia, derived as previously described (*Di Giorgio et al., 2008*). Glial cells were first plated on PDL/laminin-coated plates or coverslips in glia media composed of MEM (Life Technologies) supplemented with Glutamax (0.6%), D-glucose, and 10% horse serum (Life Technologies). After 5–7 days, neurons were thawed (Day 5 post-induction) and plated, at a density of 20,000/cm$^2$, directly onto the monolayer of mouse glia in Neurobasal medium (NBM), supplemented with NEAA, Glutamax, N2 and B27 (Life Technologies) containing BDNF (10 ng/mL, R and D systems), 2% Hyclone FBS, doxycycline (3 µg/ml), and ROCK inhibitor. Half of the media was replaced the next day and then every other day thereafter with NBM supplemented with NEAA, Glutamax, N2 and B27 containing BDNF (10 ng/mL), 2% Hyclone FBS and doxycycline (2 µg/ml).

## Immunocytochemistry

iPSCs and neurons were plated on Matrigel or PDL/laminin-coated glass coverslips, respectively. iPSCs were fixed with 4% formaldehyde (Sigma) in 4% sucrose/PBS for 10 min at room temperature and permeabilized overnight in 0.4% PBST at room temperature. iPSCs were blocked in PBS containing 0.1% triton (PBST) and 10% normal donkey serum (NDS; Jackson Immuno Research) for 1 hr at room temperature then incubated in primary antibodies for 24 hr at 4°C. The following primary antibodies were used with iPSC: SSEA4 (DSHB, MC-813–70, 3 µg/ml), Nanog (R and D, AF1997, 1:500). iPSC were washed three times with PBS then incubated for 45 min with secondary antibodies in 0.1% PBST with 2% NDS. The secondary antibodies used were Alexa 488 donkey anti-mouse and Alexa 647 donkey anti-goat (Thermo Fisher Scientific, 1:500). After secondary antibody, coverslips were washed three times with PBS, incubated for 10 min with DAPI (Invitrogen, #33342; 1:1000) and mounted onto microscope slides with Fluoromount-g (Southern Biotech).

Neurons plated onto coverslips were fixed with 3.7% formaldehyde in 4% sucrose/PBS for 15 min at room temperature and then washed three times with cold PBS. Cells were permeabilized and blocked simultaneously in 0.1% PBST with 5% normal goat serum (NGS) for 1 hr at room temperature followed by incubation with primary antibodies overnight at 4°C. The following primary antibodies were used: GFP (Abcam ab13970, AB_300798, 1:10,000), Map2 (Millipore MAB3418, AB_94856, 1:1000) and vGLUT1 (Synaptic systems, 135 302, AB_887877, 1:200). The following day coverslips were washed three times with cold PBS then incubated with secondary antibody for 1 hr at room temperature. The following secondary antibodies were used: Alexa 488 goat anti-chicken, Alexa 568 goat anti-mouse and Alexa 647 goat anti-rabbit (Thermo Fisher Scientific, 1:1000). Primary and secondary antibodies were diluted in PBS containing 5% normal goat serum. Cells were washed three

times in PBS and then briefly in distilled water and mounted onto microscope slides using ProLong Gold anti-fade reagent (Life Technologies). Neurons from the same differentiation experiment were fixed and stained at the same time with identical antibody dilutions. Images were acquired at 10x or 20x for MAP2/GFP/vGLUT1 positive neuron counting using a Leica inverted Ti microscope.

## RNA isolation and qRT-PCR

Cells were harvested by scraping from six-well plates at the indicated time points after induction of neuronal differentiation. Cells were resuspended in TRIzol Reagent (Life Technologies), and RNA was isolated following manufacturer's protocol. First-strand cDNA was synthesized from 1.1 µg of DNase I (Invitrogen) treated RNA using SuperScript IV reverse transcriptase (Thermo) and oligo dT primers following manufacturer's instructions. First-strand cDNA samples were diluted 1:7 and 2.5 µl was used in reactions. RT-PCR was performed using SYBR green on the CFX system (Bio-Rad). PCR conditions were 95°C for 3 min, then 40 cycles of 95°C for 10 s and 60°C for 30 s and a final melt curve step of from 65°C to 95°C in increments of 0.5°C per 5 s. All assays were performed in duplicate. The averaged cycle of threshold (Ct) value of two housekeeping genes (*GPI/GAPDH*) was subtracted from the Ct value of the gene of interest to obtain the ΔCt. Relative gene expression was determined as $2^{-\Delta Ct}$ (ΔΔCt) and expressed relative to the indicated sample in the experiment. Pooled human fetal brain total RNA (#636526, Lot #1612396A) and adult cortex total RNA (#636561, Lot #2007106) samples were purchased from TakaRa Bio. All primers sequences are listed in *Supplementary file 4*.

## Multi-electrode array recordings

For multielectrode array (MEA) studies, 12-well MEA plates with 64 electrodes per well were coated with PDL and laminin according to Axion Biosystems protocols. Mouse glial cells were seeded at a density of 50,000 cells/well then 30–35,000 neurons/well were added 1 week later. Every other day (Mon, Wed, Fri), half of the media was removed from each well and replaced with fresh media > 5 hr before recordings were made on those days. Independent differentiations were always plated on the same day of the week thus each time point represents the same day in the feeding schedule. Data presented in *Figures 2* and *6* are from the days on which media was changed. Spontaneous activity was recorded using Axion Biosystems Maestro 768 channel amplifier and Axion Integrated Studios (AxIS) v2.5 software. The amplifier recorded from all channels simultaneously using a gain of 1200x and a sampling rate of 12.5 kHz/channel. After passing the signal through a Butterworth band-pass filter (300–5000 Hz), on-line spike detection (threshold = 6 x the root-mean-square of noise on each channel) was performed with the AxIS adaptive spike detector. All recordings were conducted at 37°C in 5% $CO_2$/95% $O_2$. Spontaneous network activity was recorded for 5 min each day starting on day 10 of differentiation. Starting day 10 neurons were also electrically stimulated with 20 pulses at 0.5 and 0.25 Hz after spontaneous recordings were made. This anecdotally enhances maturation and migration of neurons to the electrode field.

All data reflects well-wide averages from active electrodes, with the number of wells per condition represented by N values. Active electrodes were defined as having ≥1 spikes/min. The mean firing frequency (Hz) was calculated as the total number of spikes divided by the number of active electrodes over the recording duration (300 s). The ISI CoV was calculated as the standard deviation divided by the mean inter-spike interval on active electrodes. Bursts were detected according to a Poisson distribution. The algorithm is adaptive to the mean firing rate on each electrode according to a 'surprise' threshold, adapted from *Legéndy and Salcman, 1985*. The burst frequency (Hz) was calculated as the total number of bursts divided by the number of bursting electrodes over the recording duration (300 s). The IBI CoV was calculated as the standard deviation divided by the mean inter-burst interval on bursting electrodes. The burst % was calculated as the percentage of all spikes which occurred in bursts. The number of bursts, number of spikes per burst and burst % were used as a measure of neuronal activity as this demonstrates maturity of neuronal functional properties. MEA recordings were done with two to three independent differentiations. Acute application of 500 nM apamin (Alomone) and 20 µM paxilline (Tocris) was done on day 32 in culture (See *Figure 5D*).

## Patch clamp electrophysiology

Automated voltage-clamp recordings were performed at room temperature using a Syncropatch 768PE (Nanion Technologies, Munich, Germany) as described previously (*Vanoye et al., 2018*) except that the internal solution contained 5 mM Mg-ATP. The contribution of background currents was determined by recording before and after addition of XE991 (25 µM, TOCRIS, Minneapolis, MN). Only XE991-sensitive currents were used for analysis. Whole-cell currents were elicited from a holding potential of −80 mV using 1000 ms depolarizing pulses (from −80 mV to +40 mV in +10 mV steps every 20 secs) followed by a 300 ms step to 0 mV to analyze tail currents. Cells with seal resistance ≥0.5 GΩ and series resistance ≤20 MΩ (access resistance compensation was set to 80%) were used for analysis. Peak currents were measured 999 ms after the start of the depolarizing voltage pulse and tail currents 5 ms after changing the membrane potential to 0 mV. The time-constant of activation ($\tau$) was determined by fitting currents elicited by voltage steps between −30 mV and +40 mV (50–1000 ms after start of the voltage step) to a single exponential.

Whole-cell current-clamp recordings were made from visually identified GFP-expressing neurons using an inverted Olympus IX51 microscope equipped with a 40X objective. Recording pipettes were made of glass capillaries using a horizontal Sutter P-1000 puller yielding a 2–4 MΩ resistance pipette when filled with standard K-methyl sulfate intracellular solution containing (in mM): 120 K-MeSO$_4$, 10 KCl, 10 HEPES, 10 Na$_2$-phosphocreatine, 4 Mg-ATP, 0.4 Na$_3$-GTP, pH 7.35 adjusted with KOH; osmolality 285–290 mOsm/Kg. Neurons were continuously perfused with oxygenated aCSF bath solution (in mM): 125 NaCl, 26 NaHCO$_3$, 2.5 KCl, 1.25 NaH$_2$PO$_4$, 1 MgSO$_4$, 22 glucose, 2 CaCl$_2$, pH 7.35 at 32–35°C; osmolality 310–315 mOsm/Kg.

Current-clamp recordings were acquired using a Multiclamp 700B amplifier (Molecular Devices, USA) and digitized at 10 kHz (filtered at 3 kHz) with the neurons held at −65 mV (V$_h$). All reported potential values were corrected for the liquid junction potential, calculated to be −8.2 mV. Resting membrane potential (RMP) was measured immediately after establishing the whole-cell patch clamp configuration. Input resistance (R$_N$) was calculated as the slope of the voltage-current curve determined using 500 ms current steps from −50 pA to 30 pA in 10 pA steps. Medium (mAHP) and slow (sAHP) afterhyperpolarizations (AHPs) were measured as the difference between V$_h$ and the negative going peak and 1 s after the offset of the last current step, respectively, induced by a 50 Hz train of 25 APs evoked by 2 ms/1.2 nA current injection pulses. Single AP properties, including fast afterhyperpolarization (fAHP), were measured using direct somatic current injection ramps (10–80 pA, 500 ms). AP amplitude was calculated as the difference between V$_h$ to the peak of the first AP of the ramp protocol. AP threshold was calculated where the first derivative of the up phase of the trace equaled 5 mV/ms. Using a 1 ms sliding average, the fAHP measurement was taken when the mean first derivative of the trace reached 0.0 ± 0.5 after initial spike in each sweep. AP half-width measurements were taken at half the AP peak amplitude relative to V$_h$. Neurons meeting the following criteria were used: series resistance (R$_S$) <30 MΩ, membrane resistance (R$_N$) >200 MΩ, resting potential (V$_{rest}$) < −45 mV, and AP amplitude >80 mV from V$_h$. Data were analyzed using custom MATLAB protocols (*Simkin et al., 2015*). All MATLAB scripts are available for download at github.com/sim-kind/Patch-clamp-analysis.git (*Simkin, 2021*; copy archived at swh:1:rev:bde5c7399d9f7c789fee-c0ee26ab5dad4a661d90). Data collected at three time points in culture defined as week 3 (days 14–16), week 4 (days 22–26), and week 5 (days 32–35; *Figure 3A*). Data collected from each time-point of 3–5 days were combined for statistical analysis using Statview software.

We tested the action of chronic and acute application of 1 and 20 µM XE991 (Abcam; expected to block 50% and 100% of M-current, respectively; *Wang et al., 1998*). XE991 (1 µM) was chronically applied to neuronal culture media starting day 12 in differentiation (right before beginning of week three time point) and AP properties were measured on week 4. Acute application of 20 µM XE991 or 500 nM apamin (Alomone) was done during week 4 and AP/AHP properties were measured before and 10 min after continuous perfusion of aCSF with XE991 or apamin.

## Drugs

Drugs were prepared as stock solutions using distilled water or DMSO, and then diluted to the required concentration in aCSF or culture media immediately before use. Bath-applied drugs were perfused for at least 10 min to ensure complete equilibration within the recording chamber before recording.

## Statistical analysis

Differences were evaluated using t-test, one-way or two-way ANOVA, repeated-measures ANOVA, and Fisher's protected least significant difference posthoc tests where appropriate. All data are reported as means ± SEM.

## Study approval

Written informed consent was received from participants prior to inclusion in the study under protocols approved both by Ann and Robert H. Lurie Children's Hospital of Chicago and Northwestern University IRB (#2015–738).

## Acknowledgements

We are grateful to the following funding sources: US National Institutes of Health (NIH) National Institute on Neurological Disorders and Stroke (NINDS) U54NS108874 (ALG and EK), and the New York Stem Cell Foundation (EK). We would like to thank Shoai Hattori for providing and adjusting MATLAB protocols to analyze current-clamp data, Jean-Marc Luc DeKeyser for help with molecular cloning and Anastasios (Tasso) Tzingounis for critical review of the manuscript. EK is a Les Turner ALS Research Center Investigator and a New York Stem Cell Foundation – Robertson Investigator. This work was supported, in part, by the Stanley Manne Children's Research Institute and Ann and Robert H Lurie Children's Hospital of Chicago under the Precision Medicine Strategic Research Initiative, and a gift from Davee Foundation.

## Additional information

### Funding

| Funder | Grant reference number | Author |
|---|---|---|
| NIH Office of the Director | U54NS108874 | Alfred L George Jr Evangelos Kiskinis |
| New York Stem Cell Foundation | New York Stem Cell Foundation Robertson Investigator | Evangelos Kiskinis |
| Davee Foundation | | Alfred L George |

The funders had no role in study design, data collection and interpretation, or the decision to submit the work for publication.

### Author contributions

Dina Simkin, Conceptualization, Data curation, Formal analysis, Validation, Investigation, Visualization, Methodology, Writing - original draft, Writing - review and editing; Kelly A Marshall, Reshma R Desai, Brandon N Piyevsky, Juan A Ortega, Marc Forrest, Gabriella L Robertson, Formal analysis, Investigation, Writing - review and editing; Carlos G Vanoye, Formal analysis, Investigation, Methodology, Writing - review and editing; Bernabe I Bustos, Data curation, Formal analysis, Investigation, Methodology, Writing - review and editing; Peter Penzes, Supervision, Writing - review and editing; Linda C Laux, John J Millichap, Conceptualization, Writing - review and editing, Patient recruitment; Steven J Lubbe, Formal analysis, Supervision, Writing - review and editing; Alfred L George Jr, Evangelos Kiskinis, Conceptualization, Resources, Supervision, Funding acquisition, Investigation, Visualization, Writing - original draft, Project administration, Writing - review and editing

### Author ORCIDs

Dina Simkin https://orcid.org/0000-0002-4473-5960
Carlos G Vanoye https://orcid.org/0000-0002-4935-1122
Steven J Lubbe http://orcid.org/0000-0002-7103-6671
Alfred L George Jr https://orcid.org/0000-0002-3993-966X
Evangelos Kiskinis https://orcid.org/0000-0001-8342-8616

## Ethics

Human subjects: Written informed consent was received from participants prior to inclusion in the study under protocols approved both by Ann & Robert H. Lurie Children's Hospital of Chicago and Northwestern University IRB (#2015-738).

## Decision letter and Author response

Decision letter https://doi.org/10.7554/eLife.64434.sa1
Author response https://doi.org/10.7554/eLife.64434.sa2

## Additional files

### Supplementary files

• Supplementary file 1. Single guide RNA and donor ssODN sequences for isoQ2-04$^{+/+}$, related to *Figure 1* and *Figure 1—figure supplements 2* and *3*.

• Supplementary file 2. CRISPR off-targets primers for isoQ2-04$^{+/+}$, related to *Figure 1* and *Figure 1—figure supplements 2* and *3*.

• Supplementary file 3. Whole genome sequencing comparison of Q2-04$^{R581Q/+}$ and isoQ2-04$^{+/+}$ variants.

• Supplementary file 4. Human-specific RT-qPCR primers with splice isoform specificity, related to *Figures 1*, *4* and *6*.

• Transparent reporting form

### Data availability

All data generated or analysed during this study are included in the manuscript and supporting files. Source data files have been provided for all figures.

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
