## [Decision Letter]

**Acceptance summary:**

This study provides an outstanding example of how a disease mutation in a single channel gene not only affects the function of mutant channel but also alters the functional expression of other channels leading to an unexpected changes in firing pattern. They find that these IPSC derived neurons not only have reduced KCNQ2/3 currents but show increased expression of BK and SK currents which together account for altered firing patterns. Overall, the highlights the dyshomeostatic modulation of excitability as a consequence of disease mutation in an ion channel gene.

**Decision letter after peer review:**

Thank you for submitting your article "Dyshomeostatic Modulation of Ca^2+^-Activated K^+^ Channels in a Human Neuronal Model of KCNQ2 Encephalopathy" for consideration by *eLife*. Your article has been reviewed by three peer reviewers, and the evaluation has been overseen by a Reviewing Editor and John Huguenard as the Senior Editor. The following individual involved in review of your submission has agreed to reveal their identity: Bruce P Bean (Reviewer #2).

The reviewers have discussed the reviews with one another and the Reviewing Editor has drafted this decision to help you prepare a revised submission.

Summary:

This manuscript presents a very interesting analysis of the electrophysiological consequences of a *KCNQ2* mutation associated with epilepsy, using iPSC-derived cortical excitatory neurons in culture. The authors find counter-intuitive effects of reduced KCNQ2/3 current on action potential shape and burst-firing and provide evidence that the neurons have increased BK and SK currents and that these increases can account for the altered firing. Very nicely, they show that chronic M-current inhibition can produce very similar alterations.

Overall the manuscript provides convincing evidence that reduced KCNQ2/3 current leads to unexpected changes in expression of other channels, a surprising and unexpected result. The results are of interest both for the general understanding of the limits of homeostatic changes in neuronal excitability and for the clinical manifestations, and potentially treatment, of the epilepsies produced by these mutations. In general, the experiments were well-done and with excellent controls.

Revisions:

1) The discussion and interpretation is focused exclusively on the firing properties of individual neurons. But the key feature of epilepsy is the synchronized firing of groups of neurons connected by excitatory synaptic connections. In the experimental system, it is clear from Figure 2B that the burst firing of the neurons tends to occur all at once in many neurons, presumably reflecting synaptic connections between the neurons. Some discussion of this point is needed. What happens to firing in the presence of synaptic blockers? One wonders especially about NMDA components of synaptic transmission, which can produce long lasting synaptic depolarizations to enhance burst firing of the post-synaptic neuron, while burst firing of a presynaptic neuron can in turn enhance a larger NMDA component of the post-synaptic neuron.

2) The situation in the cultures is very different than in the cortex in that the neurons are apparently all excitatory, while in the cortex GABAergic neurons are often critical in the synchronization of firing during epileptic activity. The lack of GABAergic neurons would seem to dramatically enhance positive feedback of firing of the neurons in the cultures. In fact, one wonders why the cultures of control neurons aren't wildly "epileptic", as cortical slice firing would be if GABAergic transmission is blocked.

This is an important difference than the classic studies of firing homeostasis in cultures of rodent cortical neurons by Turrigiano and colleagues, where both glutamatergic and GABAergic neurons are present. Again, I think the experimental approach stands on its own as a start, and one can guess that trying to add GABAergic neurons into the cultures would be difficult to do in a simple way that would be reproducible from culture to culture. However some discussion of this point seems called for.

3) Related to both these points, it would obviously be very interesting to know whether the changes in firing seen in the iPSC-derived neurons would also be seen in mouse models of the mutations where the channels and neurons are in a more natural context of functioning circuits (see also PMID: 28575529). Is there is any information relevant to this?

4) The idea that M-current inhibition by XE991 generates homeostatic changes (within hours) in excitability was previously put forward by Lezmy et al. (PMID: 29109270) in cultured hippocampal neurons. Besides showing this in the context of a NEE model, what is the conceptual advancement of these findings? I feel that, mechanistically, a stronger case (either by quoting the literature or by a broader Kv screening) needs to be made for why this Q2-04^R581Q/+^ model leads to hyperexcitability through altered expression patterns in other potassium channels. Along this line, why not look into other Kv channels such as the Kv1.1 and Kv1.2, which are prominently found at the distal AIS? A quick pharmacological screening would solve this important issue.

5) Although the authors nicely showed the presence of the heterozygous mutation in the patient-derived iPSCs (Q2-04^R581Q/+)^ cells, a functional expression/characterization of M-like current in those neurons was never shown. Indeed, this is also true for the isogenic Q2-04^+/+^. Please comment.

6) To determine the functional effects of R581Q, the authors heterologously expressed this variant in CHO cells either alone or with wt KCNQ3 and wt *KCNQ2* subunits. However, the unpaired nature of the experiments makes the data interpretation hard, particularly in regard to current density (even when normalized to cell capacitance). As the authors correctly state, R581Q might preclude channel conductance by disrupting tetramerization and trafficking of the channel complex in neurons. Why not assess the functional properties directly on KCNQ2-04^R581Q/+^ neurons? A comparison of the GVs, the kinetics of activation and deactivation between wt and mutant-bearing neurons would help.

7) The authors correctly state: "The intrinsic AP properties of Q 04^R581Q/+^ neurons are not consistent with a pure loss of M-current…" I wonder why the authors used XE991 to "block" a current that is either very small or absent in these cells. Or is XE991 blocking other conductance?

8) Likewise: "…but they progressively develop faster repolarization…" Although this is true at week 5 for KCNQ2-04^R581Q/+^ neurons, at week 4 these parameters are not significantly different. Furthermore, it is hard to predict what would happen with KCNQ2-04^R581Q/+^ neurons at longer times, especially because the isogenic line already started exhibiting a slower AP repolarization (or at least plateauing) at week 5. Would the same happen for KCNQ2-04^R581Q/+^ neurons?

9) Along the same subject: As shown in Figure 2, Q 04^R581Q/+^ neurons are more excitable than the wt. Figure 3D, left panel and “KCNQ2-NEE Neurons Exhibit Enhanced AP Repolarization and Post-Burst AHP” shown no difference in AP thresholds. Why? This seems to be inconsistent with the other panels in the Figure 3D. In general Figure 3D shows abrupt changes of these parameters. I also feel that Figure 6—figure supplement 1A fits better in Figure 3.

10) It is concerning that only one clone of the corrected line was used throughout the work. Similarly, while multiple control lines were used to count the number of MAP2+ neurons were generated, in MEA and patch-clamp experiments, it seemed only one mutant line and its isogenic corrected line were used. Please discuss the reasons for choosing a single clone or provide some validation data using additional clones.

---

## [Author Response]

Revisions:1) The discussion and interpretation is focused exclusively on the firing properties of individual neurons. But the key feature of epilepsy is the synchronized firing of groups of neurons connected by excitatory synaptic connections. In the experimental system, it is clear from Figure 2B that the burst firing of the neurons tends to occur all at once in many neurons, presumably reflecting synaptic connections between the neurons. Some discussion of this point is needed. What happens to firing in the presence of synaptic blockers? One wonders especially about NMDA components of synaptic transmission, which can produce long lasting synaptic depolarizations to enhance burst firing of the post-synaptic neuron, while burst firing of a presynaptic neuron can in turn enhance a larger NMDA component of the post-synaptic neuron.

The reviewers make an excellent point. To address this, in new analysis, we assessed several synchrony parameters from the MEA experiments we conducted including: “synchrony index” (Paiva et al., 2010), “Kreuz SPIKE distance” (Kreuz et al., 2013), and the “area under normalized cross-correlation”, which uses frequency domain methods to compute the cross-correlogram of all unique pair-wise combinations of electrodes in a well (Halliday et al., 2006). We present these new data to the reviewers in Author response image 1. Collectively these analyses suggest that there is an enhancement of synchrony in patient neurons relative to controls, particularly at later time points, where there is likely enhanced synaptic connectivity. However, we are hesitant to draw strong conclusions from these data, as neurons are plated in a dish, in the absence of a defined neural circuit. The question on the effects of synaptic blockers and NMDA is an intriguing one and we plan on pursuing these experiments in the future.

**Author response image 1. respfig1:** Synchrony metrics assessed by MEA experiments.

2) The situation in the cultures is very different than in the cortex in that the neurons are apparently all excitatory, while in the cortex GABAergic neurons are often critical in the synchronization of firing during epileptic activity. The lack of GABAergic neurons would seem to dramatically enhance positive feedback of firing of the neurons in the cultures. In fact, one wonders why the cultures of control neurons aren't wildly "epileptic", as cortical slice firing would be if GABAergic transmission is blocked.This is an important difference than the classic studies of firing homeostasis in cultures of rodent cortical neurons by Turrigiano and colleagues, where both glutamatergic and GABAergic neurons are present. Again, I think the experimental approach stands on its own as a start, and one can guess that trying to add GABAergic neurons into the cultures would be difficult to do in a simple way that would be reproducible from culture to culture. However some discussion of this point seems called for.

The reviewers make a very good point. Our approach has been reductionist. The goal of this study is to characterize the effects of a *KCNQ2* mutation in excitatory neurons cultured alone, which as the reviewers point out “is a good start”. We are currently building more sophisticated model systems by integrating excitatory and inhibitory neurons to assess their interactions and the impact of *KCNQ2* mutations. It will be fascinating to examine the homeostatic changes that occur when both of these neuronal subtypes are present. We discuss this in the revised Discussion.

3) Related to both these points, it would obviously be very interesting to know whether the changes in firing seen in the iPSC-derived neurons would also be seen in mouse models of the mutations where the channels and neurons are in a more natural context of functioning circuits (see also PMID: 28575529). Is there is any information relevant to this?

We are planning to address this point experimentally by examining the physiological properties of early postnatal neurons in mutant *KCNQ2* mouse models in the future. Interestingly, Biba et al., (bioRxiv, https://doi.org/10.1101/2020.05.12.090464) recently reported that pyramidal neurons in heterozygous knock-in mice harboring the loss-of-function pathogenic T274M variant, exhibited hyperexcitability early on (P7-P9), but this effect went away later (P30) (Biba et al., 2020). While the involvement of other channels, or the AHP specifically, were not examined, these results suggest that homeostatic changes also take place in neurons in vivo in response to *KCNQ2* mutations. We discuss this point in the revised Discussion.

4) The idea that M-current inhibition by XE991 generates homeostatic changes (within hours) in excitability was previously put forward by Lezmy et al. (PMID: 29109270) in cultured hippocampal neurons. Besides showing this in the context of a NEE model, what is the conceptual advancement of these findings? I feel that, mechanistically, a stronger case (either by quoting the literature or by a broader Kv screening) needs to be made for why this Q2-04^R581Q/+^ model leads to hyperexcitability through altered expression patterns in other potassium channels. Along this line, why not look into other Kv channels such as the Kv1.1 and Kv1.2, which are prominently found at the distal AIS? A quick pharmacological screening would solve this important issue.

We chose to focus on Ca^2+^-activated K^+^ channels specifically due to their roles in the post-burst AHP that we determined to be paradoxically enhanced in patient neurons. Nevertheless, the reviewers make a valid point about interrogating other Kv channels. To address their concern, in new data we assayed gene expression changes in several ion channels in addition to BK/IK/SKs, including KCND2, KCNT1, KCNA1, KCNA2, KCNA4, KCNQ2, KCNQ3, KCNQ5, HCN1, HCN2, SCN8A and ANKG (known to bind KCNQ2 in the AIS). We find that compared to controls, Q2-04^R581Q/+^ neurons exhibited lower *KCNA4* and *KCNQ5* expression on week 3 and 5 respectively, and increased *KCNA1* and *KCNA2* expression on week 5. The expression level of all other genes examined was unaffected. The upregulation of *KCNA1* and *KCNA2* is an interesting new finding as both genes are associated with epilepsy and could be contributing to enhanced AP repolarization that we find at week 5. Thus, we cannot rule out other maladaptive changes besides SK and BK channel up-regulation in our study. We present these data in new Figure 4—figure supplement 1 and include discussion on this point. In addition, we discuss the novelty of our findings in the context of the important study of Lezmy et al. in the revised Discussion.

5) Although the authors nicely showed the presence of the heterozygous mutation in the patient-derived iPSCs (Q2-04^R581Q/+)^ cells, a functional expression/characterization of M-like current in those neurons was never shown. Indeed, this is also true for the isogenic Q2-04^+/+^. Please comment.

We previously attempted to measure M-current using whole-cell voltage-clamp recording but discovered that there was excessive current run down that was variable cell-to-cell. This confounded our ability to reliably interpret these measurements. We plan to try perforated patch recording but we won’t have data available in the foreseeable future.

6) To determine the functional effects of R581Q, the authors heterologously expressed this variant in CHO cells either alone or with wt KCNQ3 and wt KCNQ2 subunits. However, the unpaired nature of the experiments makes the data interpretation hard, particularly in regard to current density (even when normalized to cell capacitance). As the authors correctly state, R581Q might preclude channel conductance by disrupting tetramerization and trafficking of the channel complex in neurons. Why not assess the functional properties directly on KCNQ2-04^R581Q/+^ neurons? A comparison of the GVs, the kinetics of activation and deactivation between wt and mutant-bearing neurons would help.

As explained by our response to point #5, we are unable to reliably record channel properties in the iPSC-neurons. However, we did determine the gating kinetics and voltage-dependence of activation for heterologous expressed WT and mutant KCNQ2, and we present these new data to a revised Figure 1—figure supplement 1.

7) The authors correctly state: "The intrinsic AP properties of Q 04^R581Q/+^ neurons are not consistent with a pure loss of M-current…" I wonder why the authors used XE991 to "block" a current that is either very small or absent in these cells. Or is XE991 blocking other conductance?

The effects of XE991 have been studied for over 20 years and it is widely regarded as a highly selective M-current inhibitor (Jentsch, 2000; Wang et al., 2000; Robbins, 2001; Greene et al., 2017). However, there are reports showing that when used at concentrations of 10µM or higher, it can have very small effects on other (cardiac) K^+^ channels (Elmedyb et al., 2007; Zhong et al., 2010). While we cannot exclude the possibility that XE991 blocks other conductances during the acute treatments we performed (20 μM, Figure 3—figure supplement 1C-F), it is highly unlikely that this was the case for chronically treating neurons (1µM, Figure 6), ensuring that the maladaptive compensatory changes we observed were driven by M-current inhibition.

8) Likewise: "…but they progressively develop faster repolarization…" Although this is true at week 5 for KCNQ2-04^R581Q/+^ neurons, at week 4 these parameters are not significantly different. Furthermore, it is hard to predict what would happen with KCNQ2-04^R581Q/+^ neurons at longer times, especially because the isogenic line already started exhibiting a slower AP repolarization (or at least plateauing) at week 5. Would the same happen for KCNQ2-04^R581Q/+^ neurons?

We refer to both AP half-width and fAHP as measurements of AP repolarization. Thus Q2-04^R581Q/+^ neurons start out having slower AP repolarization compared to isoQ2-04^+/+^ at week 3 (due to slower AP half-width) but by week 4, while the AP half-width is not yet different, Q2-04^R581Q/+^ neurons exhibit a significantly larger fAHP. By week 5, Q2-04^R581Q/+^ neurons exhibit both faster AP half-widths and larger fAHP indicated enhanced AP repolarization. To further clarify this point we revised the text: “However, our analysis of Q2-04^R581Q/+^ neurons showed that during week 3, AP repolarization was slower (with longer AP half-width) compared to isoQ2-04^+/+^ but over time AP repolarization became more pronounced with faster AP half-width and enhanced fAHP.”

We agree with the reviewers that it will be interesting to assess these parameters at longer times but it is hard to predict what would happen. The longer these cells are kept in culture the harder it is to maintain their health thus we avoided doing recordings past week 5.

9) Along the same subject: As shown in Figure 2, Q 04^R581Q/+^ neurons are more excitable than the wt. Figure 3D, left panel and “KCNQ2-NEE Neurons Exhibit Enhanced AP Repolarization and Post-Burst AHP” shown no difference in AP thresholds. Why? This seems to be inconsistent with the other panels in the Figure 3D. In general Figure 3D shows abrupt changes of these parameters. I also feel that Figure 6—figure supplement 1A fits better in Figure 3.

Figure 2 does not specifically show that Q2-04^R581Q/+^ neurons have more activity than controls but rather that their activity is limited to burst firing. While AP threshold may have some effect on spike initiation in a burst, faster AP repolarization allows the neuron to reset voltage-gated Na^+^ channels and be able to fire again repeatedly. Furthermore, different conductances are responsible for AP threshold as compared to AP repolarization. Specifically, BK and SK channels that we show to be upregulated here, do not play a significant role in AP threshold.

Our reasoning for placing Figure 6—figure supplement 1 separately from Figure 3. is the following: Figure 6—figure supplement 1A is the supplement for main Figure 6 and compares the passive properties (RMP and IR) of isoQ2-04^+/+^ chronically treated with XE991 to untreated neurons only on week 4. Figure 3 is a time course of active membrane properties on weeks 3, 4 and 5.

10) It is concerning that only one clone of the corrected line was used throughout the work. Similarly, while multiple control lines were used to count the number of MAP2+ neurons were generated, in MEA and patch-clamp experiments, it seemed only one mutant line and its isogenic corrected line were used. Please discuss the reasons for choosing a single clone or provide some validation data using additional clones.

The selection of a single patient and isogenic corrected clone for analysis is based on our experience in developing iPSC-based models and was made after a series of quality control steps. The iPSC lines were generated by a non-integrating reprograming method, thus excluding the possibility of random genetic integrations affecting individual clones. The clones selected for these experiments expressed pluripotency markers, exhibited normal karyotypes and were assessed by both targeted Sanger sequencing, and whole genome sequencing. In addition, in a separate line of investigation, we have interrogated the electrophysiological pattern of neurons derived from two independent patient clones in relation to two independent isogenic corrected clones for this individual, and have found that they behave similarly.